# The glycosylation design space for recombinant lysosomal replacement enzymes produced in CHO cells

Weihua Tian [1], Zilu Ye[1], Shengjun Wang[1], Morten Alder Schulz[1], Julie Van Coillie [1], Lingbo Sun[1], Yen-Hsi Chen[1], Yoshiki Narimatsu[1], Lars Hansen[1], Claus Kristensen[2], Ulla Mandel[1], Eric Paul Bennett[1], Siamak Jabbarzadeh-Tabrizi[3], Raphael Schiffmann[3], Jin-Song Shen[3], Sergey Y. Vakhrushev[1], Henrik Clausen [1] & Zhang Yang [1]

Lysosomal replacement enzymes are essential therapeutic options for rare congenital lysosomal enzyme deficiencies, but enzymes in clinical use are only partially effective due to short circulatory half-life and inefficient biodistribution. Replacement enzymes are primarily taken up by cell surface glycan receptors, and glycan structures influence uptake, biodistribution, and circulation time. It has not been possible to design and systematically study effects of different glycan features. Here we present a comprehensive gene engineering screen in Chinese hamster ovary cells that enables production of lysosomal enzymes with N-glycans custom designed to affect key glycan features guiding cellular uptake and circulation. We demonstrate distinct circulation time and organ distribution of selected glycoforms of α-galactosidase A in a Fabry disease mouse model, and find that an α2-3 sialylated glycoform designed to eliminate uptake by the mannose 6-phosphate and mannose receptors exhibits improved circulation time and targeting to hard-to-reach organs such as heart. The developed design matrix and engineered CHO cell lines enables systematic studies towards improving enzyme replacement therapeutics.

[1] Copenhagen Center for Glycomics, Departments of Cellular and Molecular Medicine and Odontology, Faculty of Health Sciences, University of Copenhagen, Blegdamsvej 3, 2200 N Copenhagen, Denmark. [2] GlycoDisplay ApS, Blegdamsvej 3, 2200 Copenhagen N, Denmark. [3] Institute of Metabolic Disease, Baylor Scott & White Research Institute, 3812 Elm Street, Dallas, TX 75226, USA. Correspondence and requests for materials should be addressed to H.C. (email: hclau@sund.ku.dk) or to Z.Y. (email: yang@sund.ku.dk)

Lysosomal storage diseases (LSDs) are characterized by the progressive accumulation of undegraded metabolites that lead to lysosomal and cellular dysfunction[1,2]. A variety of therapeutic approaches have been developed for LSDs, with intravenous enzyme replacement therapies (ERTs) being the most prevalent[3], but ERTs for LSDs still face major challenges, and the most important may be the delivery of the infused recombinant enzymes to hard-to-reach organs, such as bone, kidney, heart, and brain[4]. Lysosomal enzymes are glycoproteins and the cellular uptake of replacement enzymes is thought to primarily rely on cell-surface receptors recognizing N-glycan features[5], including the mannose 6-phosphate (M6P) receptors (MPRs)[6], Ashwell–Morell receptor (AMR) (asialoglycoprotein receptor)[7], and mannose receptor (MR)[8], and these participate in cellular uptake and lysosomal delivery of therapeutic glycoproteins. The MPRs specifically recognize terminal M6P attached to high-mannose and hybrid-type N-glycans, and they direct both intracellular delivery of lysosomal enzymes as well as uptake of exogenous M6P-containing glycoproteins from circulation[6]. The AMR expressed primarily on liver hepatocytes recognizes glyco-proteins with uncapped terminal galactose (Gal) or N-acetyl-galactosamine (GalNAc) residues, and mediates clearance from the circulation[7]. The MR expressed primarily on mononuclear macrophages binds mainly exposed mannose (Man), N-acetyl-glucosamine (GlcNAc) and fucose (Fuc) residues on N-glycans, and directs uptake of glycoproteins and targeting to endosomes and lysosomes[8]. Tissue distribution and circulation time of infused replacement enzymes are at least partly dependent on the expression of these receptors[9]. Other glycan-binding proteins including Siglecs and Galectins may bind therapeutic N-glycoproteins[10,11], and glycan-independent uptake of lysosomal enzymes by, for example, the low-density lipoprotein receptor proteins (LRPs) has been reported[12]. The glycosylation state of replacement enzymes is critical for the pharmacokinetic proper-ties and therapeutic effect, with the key determining N-glycan features being the degree of M6P tagging and exposure of term-inal Man, Gal, and/or GlcNAc residues in a complex interplay yet unexplored. Most currently approved ERTs have highly hetero-geneous N-glycan structures as dictated by the inherent glyco-sylation capacity of CHO cells[13]. However, there have been limited options for custom design of the glycosylation capacity of CHO cells and thus for testing specific ERTs with different N-glycan features to explore potential improved therapeutic performance.

Different strategies have been undertaken to explore gly-coengineering as a means to improve delivery of ERTs, including use of exoglycosidases for postproduction enzyme modification[14], as well as the use of engineered yeast[15] and plant production platforms[16,17]. Pioneering work originally demonstrated how glycosidase trimming of N-glycans on β-glucoscerebrosidase (GBA) resulted in efficient targeting to macrophages through the MR and provided a successful therapy for non-neuropathic Gaucher disease[18]. The first recombinant GBA with Man-terminated (high-Man) N-glycans was produced in CHO cells followed by postproduction exoglycosidase treatment[19], and similar GBA products are produced in human cells by use of mannosidase I inhibitor (kifunensine)[20] or in carrot cells[21]. Most strategies for glycoengineering of lysosomal enzymes have sought to improve targeting by MPRs and MRs by increasing the content of M6P or exposed Man residues[15,22–24]. However, these glyco-forms will cause rapid and efficient uptake by especially the liver and spleen, while targeting to other organs may be limited[4]. Early studies demonstrated that increased content of sialic acid (SA) on lysosomal enzymes isolated from plasma improves their circula-tion time similar to other types of therapeutic glycoproteins[25], but further studies were hampered by lack of methods to produce

these recombinantly. Oxidative degradation and reduction of glycans on enzymes has provided therapeutic efficacy with extended circulation time and wider biodistribution[26]; however, this partly inactivates the enzyme and may not be suitable for clinical production[27,28].

Most ERTs are produced in CHO cells, and with the advent of efficient precise gene editing tools, it is now possible to introduce extensive engineering designs to optimize the glycosylation capacity of CHO cells[29]. Here, we present a comprehensive screen of engineering options for lysosomal enzymes in CHO cells, and we provide a panel of glycoengineered CHO cell lines with dif-ferent capacities for producing lysosomal enzymes furnished with all the key glycan features known to affect cellular uptake and circulation time. The genetic design matrix developed makes it possible to investigate ERTs with a diverse array of glycoforms. We used the α-galactosidase A (GLA) as a representative ERT in a mouse model of Fabry disease, and demonstrate how distinct glycoforms of GLA are differentially targeted to liver, spleen, kidney, and heart, and present evidence that GLA glycoforms capped with α2-3-linked SA (α2-3SA), but surprisingly not α2-6SA, exhibit improved circulation and biodistribution. Thus, in contrast to the current dogma, α2-3SA-capped glycoforms of lysosomal enzymes may represent a strategy to overcome the most critical problems of rapid clearance in liver and poor bio-distribution found with current ERTs.

## Results

**Glycoengineering of lysosomal enzymes produced in CHO cells.** A stable wild-type (WT) CHO clone expressing human GLA was established and used for a gene knockout (KO) tar-geting screen using clustered regularly interspaced short palin-dromic repeats/CRISPR-associated protein 9 (CRISPR/Cas9) considering all glycosyltransferases and hydrolases functioning in N-glycosylation and M6P processing, as well as receptors and other proteins (Fig. 1). RNA-sequencing expression profiling was used to identify relevant genes expressed in CHO cells (Supple-mentary Fig. 1). We used site-specific glycoprofiling of the secreted purified GLA to monitor effects on glycosylation. The occupancy at individual N-glycosites was evaluated by comparing the peptide/glycopeptide ratio for each N-glycosite. GLA has three N-glycosites (N108, N161, and N184), and when expressed in CHO WT cells, GLA was site-specifically glycosylated with mainly complex structures capped with SA at N108 and with M6P-tagged high-mannose-type glycans at N161 and N184 (Fig. 2a and Supplementary Fig. 2, #1–3), in agreement with previous reports[30]. We targeted 43 genes individually or in rational combinations guided by the sequential biosynthetic pathway of N-glycans and known groups of isoenzymes with overlapping functions (Supplementary Table 2 and Supplemen-tary Data 1). Figure 1 presents a summary of the observed general trend effects of the screen for SA, M6P, and Man. In general the occupancies at the three glycosites were found to be near com-plete for GLA produced in the CHO mutants, with the exception of the N184 glycosite where KO of Alg5 and Alg6 reduced the occupancy.

Targeting the lipid-linked oligosaccharide precursor assembly on the cytosolic side (Alg1/2/11/13/14) was not successful since viable cells with bi-allelic KO could not be established in agreement with similar observations in yeast[31,32]; however, targeting the precursor assembly on the ER luminal side (Alg3/5/6/8/9/12) produced surprising options for site-specific engineering of M6P-tagging of GLA. KO of Alg3 substantially enhanced M6P tagging at N108, while reducing M6P at N161 (Fig. 2b and Supplementary Fig. 2, #4–5). KO of Alg9 reduced M6P at N161 and increased tagging at N184 (Fig. 2c and

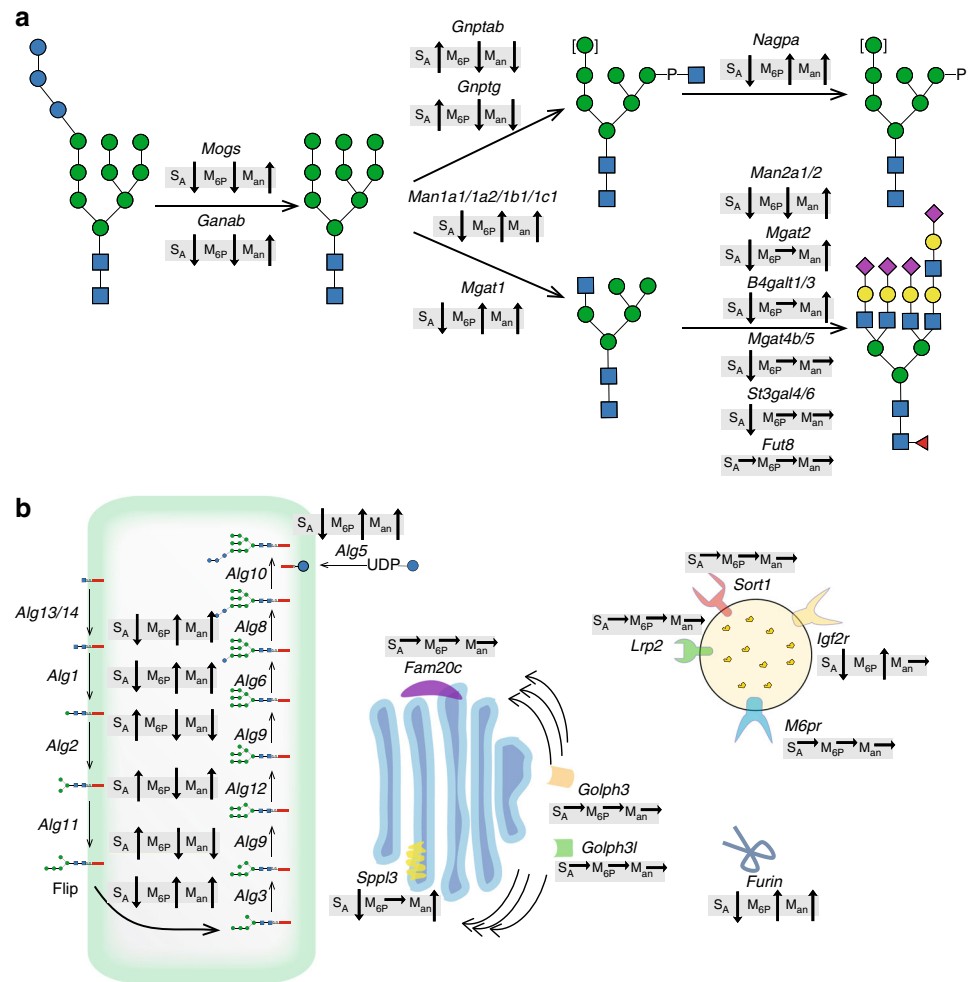

**Fig. 1** Graphic depiction of gene targeting screen performed in CHO cells with general trend effects on N-glycosylation of α-galactosidase A (GLA). clustered regularly interspaced short palindromic repeats/CRISPR-associated protein 9 (CRISPR/Cas9) knockout (KO) targeted genes are indicated with their predicted functions. **a** The general trend effects of KO targeting of glycosyltransferase, glycosylhydrolase, and other related genes known to function in N-glycosylation and mannose 6-phosphate (M6P) tagging are indicated for changes in total sialic acid capping (SA), M6P-tagging (M6P), and exposed terminal mannose (Man), with arrows indicating increase/decrease. **b** Trend effects of KO targeting of genes encoding enzymes functioning in the dolichol-linked precursor oligosaccharide assembly, receptors involved in trafficking of lysosomal enzymes, and other proteins reported to affect stability of enzymes in the Golgi. Glycan symbols according to SNFG format[70]

Supplementary Fig. 2, #6). KO of *Alg12* reduced M6P at N161 and increased M6P at N184 (Fig. 2d and Supplementary Fig. 2, #7). KO of *Alg6* and *Alg8* enhanced hybrid structures with one branch capped by SA and one with M6P at N161 (Fig. 2e, f and Supplementary Fig. 2, #9–10). KO of *cis*-Golgi mannosidases (*Man1a1/1a2/1b1/1c1*) enriched oligomannose structures and enhanced M6P at all three glycosites (Fig. 2g and Supplementary Fig. 2, #12–16). KO of medial Golgi mannosidase (*Man2a1/2*) created hybrid N-glycans with one branch capped by SA and one with oligomannose at the expense of reduced M6P (Fig. 2h and Supplementary Fig. 2, #17). KO of *Mgat1* as expected completely eliminated complex N-glycans, and interestingly enhanced M6P tagging at N161 and N184 (Fig. 2i and Supplementary Fig. 2, #18). KO of *Mgat2* produced the mono-antennary hybrid-type N-glycan at N108 without affecting M6P at N161 and N184 (Fig. 2j and Supplementary Fig. 2, #19), while KO of *Mgat4b/5* completely eliminated tri- and tetra-antennary N-glycans and increased homogeneity (Fig. 2k and Supplementary Fig. 2, #20). The results demonstrate how the content and position of M6P and exposed Man on lysosomal enzymes can be fine-tuned in great detail by gene engineering of CHO cells. Targeting the N-glycan ER glucosidases (*Mogs/Ganab*) to probe the role of the Glc

residues and chaperone interactions did not affect secretion of GLA substantially (Supplementary Fig. 3), and demonstrated that GLA glycoforms with retained Glc residues and M6P tagging can be produced (Fig. 2l, m and Supplementary Fig. 2, #21–22).

Targeting the M6P tagging process by KO of *Gnptab* or *Gnptg* of the GlcNAc-1-phosphotransferase complex enabled production of GLA with rather homogeneous complex N-glycans capped by SA at all N-glycosites, but lacking M6P residues (Fig. 2n, o and Supplementary Fig. 2, #23–24). In addition, KO of the GlcNAc-1-phosphate hydrolase (*Nagpa*) uncovering enzyme resulted in GLA with GlcNAc residues retained on M6P and interestingly increased M6P tagging, including substantial increase in bis-M6P (Fig. 2p and Supplementary Fig. 2, #26). In addition to two high-affinity M6P binding sites, the large cation-independent mannose 6-phosphate receptor (CI-MPR) contains another preferential binding site for M6P-GlcNAc[33]. Targeting the M6P-tagging process may also affect lysosomal targeting of some endogenous CHO cell lysosomal enzymes[12,34], and resulting changes in secreted lysosomal glycosylhydrolases, for example, neuraminidase 1 (*Neu1*) may affect glycan structures of recombinant expressed enzymes. KO of *B4galt1/3* reduced galactosylation and resulted in exposed GlcNAc residues primarily at N108 (Fig. 2q

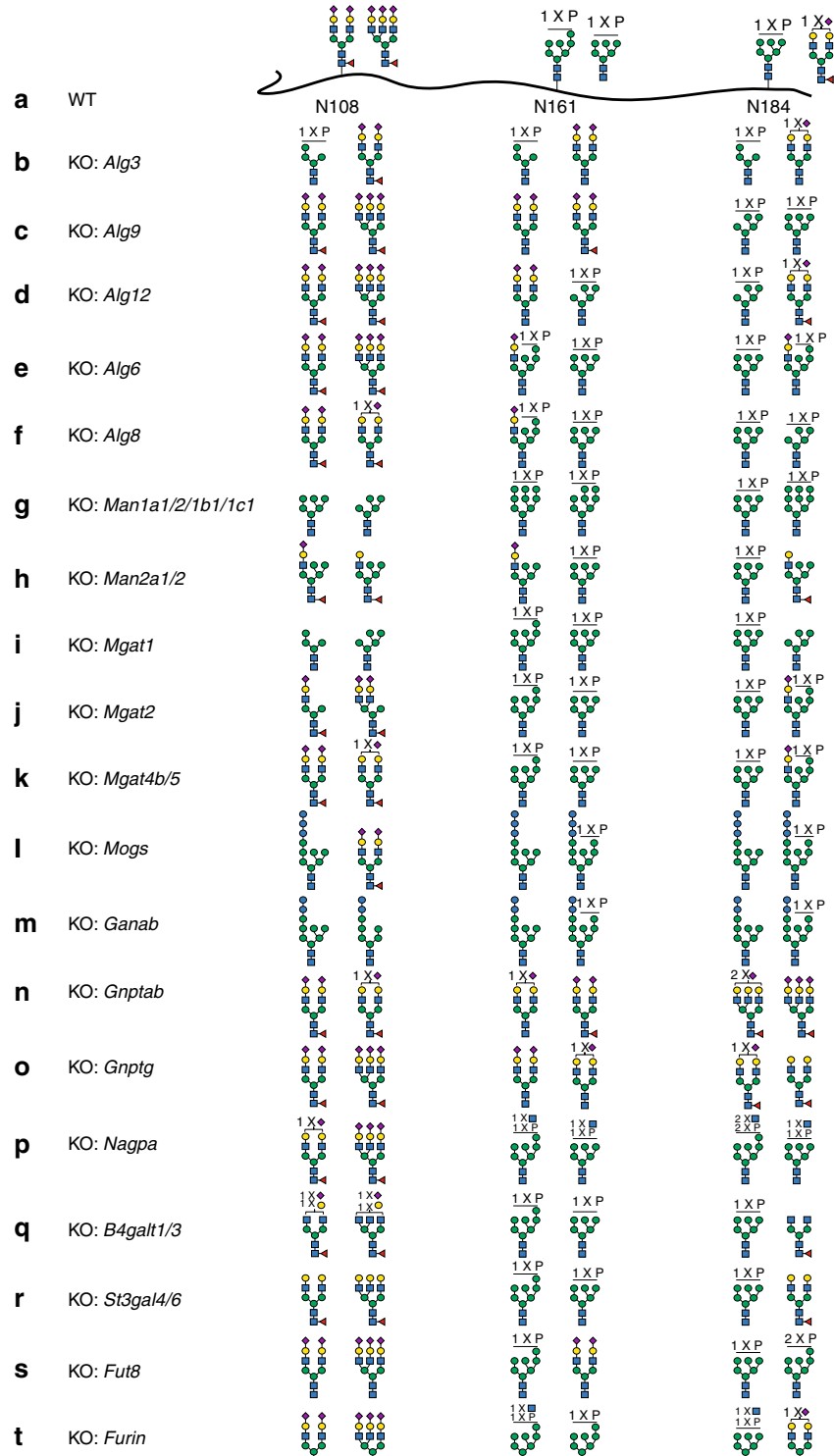

**Fig. 2** Site-specific glycan analyses of selected α-galactosidase A (GLA) glycoforms produced in the initial knockout/knock-in (KO/KI) CHO cell screen. **a** The two most abundant glycan structures at N-glycosites (N108, N161, and N184) of GLA produced in CHO wild type (WT) are shown, and in **b**–**t** the two most abundant glycans for GLA produced in engineered CHO clones are shown as indicated. The detailed N-glycan analyses of all GLA glycoforms are shown in Supplementary Fig. 2 together with additional variants. Each glycan structure was confirmed by targeted tandem mass spectrometry (MS/MS) analysis (Supplementary Fig. 5). Details regarding the stacking ancestry and sequence analysis are shown in Supplementary Table 2 and Supplementary Data 1

and Supplementary Fig. 2, #31). Targeting sialylation by KO of *St3gal4/6* substantially reduced SA capping and resulted in the exposure of terminal Gal residues (Fig. 2r and Supplementary Fig. 2, #32). Furthermore, KO of *Fut8* eliminated core fucose without affecting other features (Fig. 2s and Supplementary Fig. 2, #33).

We also targeted the genes encoding the M6P receptors CI-MPR (*Igf2r*) and CD-MPR (*M6pr*), which did not substantially affect glycosylation of the secreted GLA, although KO of *Igf2r* slightly increased bis-M6P tagging at the N184 glycosite (Supplementary Fig. 2, #34–35). Targeting the late-acting signal peptidase, *Sppl3*, shown to play a role in shedding of glycosyltransferases[35], induced a slight increase of exposed Man (Supplementary Fig. 2, #38). KO of *Furin*, important for activation of *Nagpa*[36], resulted in similar N-glycan profile with the accumulation of GlcNAc-1-P residues, as found with KO of *Nagpa* (Fig. 2t and Supplementary Fig. 2, #39). KO of phosphokinase *Fam20c* and the phosphatidylinositol-4-phosphate effector *Golph3* and Golgi protein *Golph3l* did not substantially affect the N-glycosylation of GLA (Supplementary Fig. 2, #40–42).

**Combinatorial glycoengineering**. The individual gene KO screen provides a matrix for design of combinatorial engineering to produce GLA with a wider range of desirable glycoforms. We first explored designs without M6P tagging. Stacking KO of *Gnptab/g* with KO of *Mgat1* enabled production of GLA with high-mannose N-glycans at all three glycosites (Fig. 3a and Supplementary Fig. 2, #44), which will bind MR expressed on macrophages and efficiently target the liver and spleen[19,37–40]. Stacking KO of *Man2a1/2* involved in α-mannosidase trimming generated GLA with a mono-antennary hybrid structure with a complex sialylated α3-arm combined with three Man residues on the α6-arm (Fig. 3b and Supplementary Fig. 2, #45). Stacking KO of *Mgat2* enables production of GLA with mono-antennary hybrid N-glycan (Fig. 3c and Supplementary Fig. 2, #46), and stacking KO of *Mgat4b/5* enables production of GLA with homogeneous bi-antennary complex N-glycans with SA capping (Fig. 3d and Supplementary Fig. 2, #47), which could be combined with *Mgat2* KO[29].

Next, we focused on improving M6P tagging by first testing individual knock-in (KI) of *GNPTG* or *GNPTAB*, which enhanced M6P at N161 and N184, and KI of *GNPTAB* induced bis-M6P at N184 (Fig. 3e, f and Supplementary Fig. 2, #48–51). Moreover, combined KI of both genes induced a substantial increase in M6P tagging at all three glycosites and with high content of the mono-antennary hybrid structure with SA and M6P (Fig. 3g and Supplementary Fig. 2, #52–53). KI of *GNPTAB* combined with KO of *Alg3* enabled production of a unique high-Man N-glycan with efficient M6P tagging at all N-glycosites (Fig. 3h and Supplementary Fig. 2, #54-55). The CI-MPR has multiple binding sites and has the capacity to bind diverse M6P-tagged structures with different affinities[41], and increasing the M6P content and introducing bis-M6P are predicted to enhance uptake as demonstrated, for example, with the acid α-glucosidase used for ERT of Pompe disease[23].

CHO WT cells only have capacity for α2-3SA capping, and systematic studies of the influence of α2-3SA versus α2-6SA capping found on most human serum glycoproteins have not been performed with native glycoproteins. Targeted KI of *ST6GAL1* in cells with KO of *Gnptab* and *St3gal4/6* enabled production of GLA with homogeneous α2-6SA capping (Fig. 3i and Supplementary Fig. 2, #58). Combining KO of *Mgat4b/5* with KI of *ST3GAL4* enabled production of homogenous bi-antennary N-glycans capped with α2-3SA (Fig. 3j and Supplementary Fig. 2,

#59). Combined with KO of *Fut8*, any glycoform may likely be produced without core fucose (Fig. 3k, l and Supplementary Fig. 2, #60, 61).

**The glycoengineering matrix is applicable to other ERTs**. To validate the glycoengineering designs for other lysosomal enzymes, we tested representative designs with GBA that has four N-glycans (Fig. 4 and Supplementary Fig. 4 and Supplementary Tables 3 and 4)[38]. KO of *Alg3* increased the M6P content of GBA in particular for N146 and N270 glycans (Fig. 4b and Supplementary Fig. 4, #2). KO of *Alg9* had little effect on N-glycans at N19 and N59, but altered the oligomannose structures with M6P at N146 and N270 (Fig. 4c and Supplementary Fig. 4, #3). Targeting *Gnptab* resulted in rather homogeneous complex type N-glycans with SA capping at all four N-glycosites and no M6P content (Fig. 4d and Supplementary Fig. 4, #4). Targeting *Mgat1* enabled production of GBA without complex type N-glycans, but with high-mannose glycans and reduced M6P mainly at N270 (Fig. 4e and Supplementary Fig. 4, #5–6). Stacked KO of *Gnptab* and *Mgat1* enabled production of GBA with high-mannose N-glycans (4–5 Man) without M6P at all glycosites (Fig. 4f and Supplementary Fig. 4, #7–8). Stacked KO of *Man2a1/2* and *Gnptab* generated GBA with a rather homogeneous mono-antennary hybrid structure at all four glycosites with a complex sialylated α3-arm combined with three Man residues on the α6-arm (Fig. 4g and Supplementary Fig. 4, #9). Similarly, GBA with mono-antennary hybrid N-glycans carrying a single Man residue at the α6-arm was generated by stacking KO of *Mgat2* and *Gnptab* (Fig. 4h and Supplementary Fig. 4, #10). In general, the outcome of the engineering performed with GBA correlated well with the effects observed with GLA, when considering the inherent site specificity of N-glycan processing found with the enzymes expressed in WT CHO cells.

**Analyses of GLA glycoforms in a Fabry mouse model**. Fabry disease is caused by deficiency in GLA activity, and the leading ERT is Fabrazyme (Sanofi Genzyme) produced in CHO cells[30]. We first benchmarked GLA produced in our CHO WT cell (~100 mg/L) with a clinical lot of Fabrazyme finding lower content of exposed Man residues on GLA produced in our CHO WT cells (Fig. 5a and Supplementary Fig. 2, #1, 3). The GLA enzyme produced in our CHO WT cells exhibited similar blood circulation half-time (12.0 ± 0.3 min) (mean ± standard deviation) as Fabrazyme (11.9 ± 2.3 min) (Fig. 5b), but with trends of higher liver targeting and lower spleen, kidney, and heart targeting of our GLA variant compared to Fabrazyme (4 h after infusion), although only the lower kidney targeting was significant (Fig. 5d). We chose to use Fabrazyme for comparison in further studies to enable direct comparison with results from the literature.

We then tested five distinct glycoforms of GLA (Fig. 5a and Supplementary Fig. 2, #6, 55, 52, 59, 58). The specific activity and stability in plasma of these GLA glycovariants were essentially identical (Supplementary Fig. 6). In a first experiment we used one infusion dose of 1 mg/kg (Exp. #1) (Fig. 5c, e). The three glycoforms designed with slightly lower M6P (GLA-LoM6P), higher M6P (GLA-HiM6P), or higher M6P content with mainly the hybrid type (GLA-HybM6P) produced trends towards higher or lower circulation time with half-lives of 15.4 ± 1.1, 11.0 ± 2.0, and 8.3 ± 0.8 min, respectively, compared with 9.8 ± 0.3 min for Fabrazyme (Fig. 5c). These three glycoforms showed only minor differences in targeting to select organs compared to Fabrazyme (24 h after infusion), with the LoM6P glycoform yielding significantly higher levels of enzyme activity in the heart, and the HiM6P and HybM6P glycoforms exhibiting lower levels in the spleen (Fig. 5e).

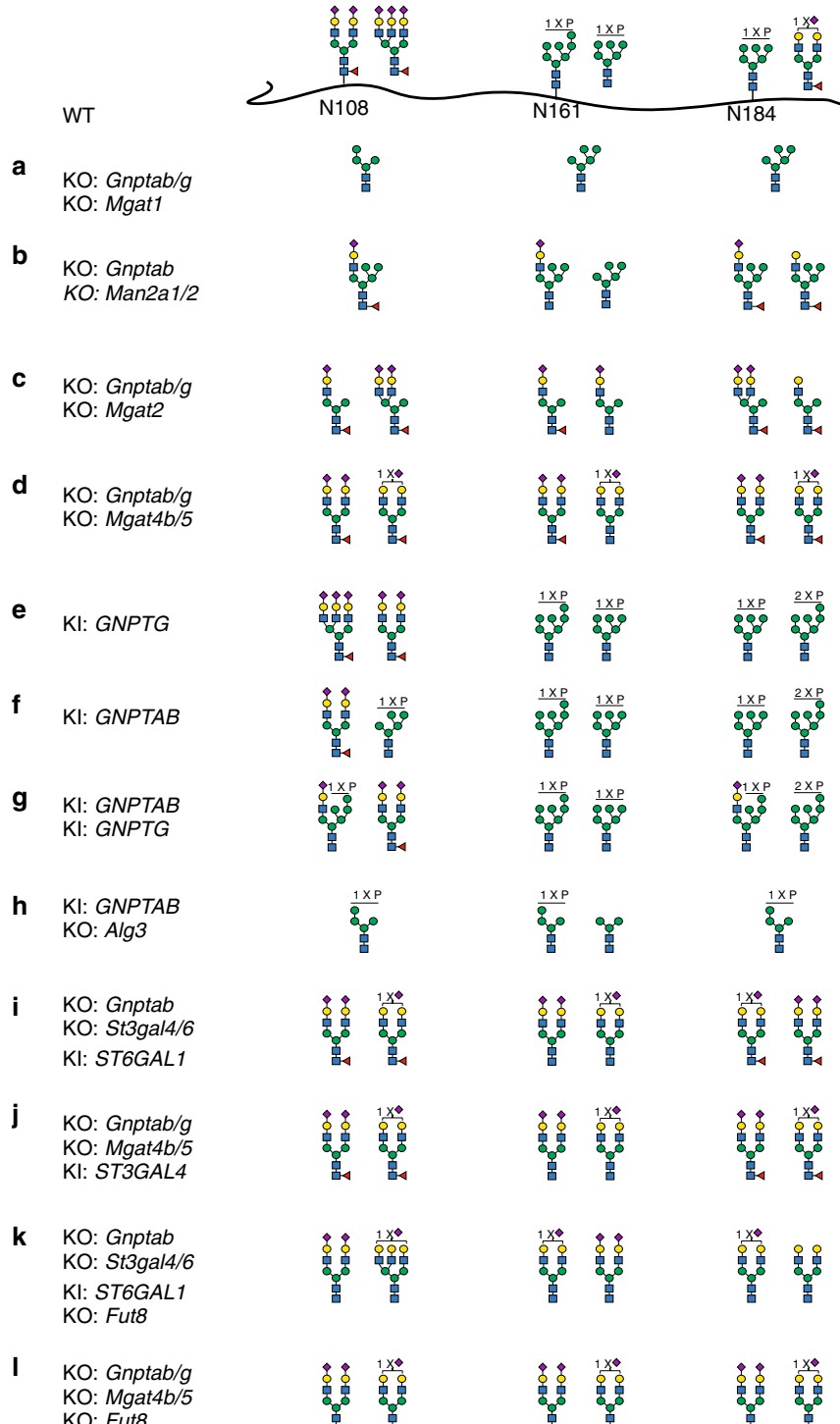

**Fig. 3** Site-specific glycan analyses of α-galactosidase A (GLA) glycoforms produced with combinatorial gene engineering of CHO cells. **a–l** The two most abundant glycan structures at N-glycosites (N108, N161, and N184) of GLA are shown. Each glycan structure was confirmed by targeted tandem mass spectrometry (MS/MS) analysis. The detailed N-glycan analyses of all GLA glycoforms are shown in Supplementary Fig. 2 together with additional variants. Details regarding the stacking ancestry and sequence analysis are shown in Supplementary Table 2 and Supplementary Data 1

In striking contrast, the two glycoforms designed with N-glycans capped by SA and without M6P and exposed Man produced significant changes in circulation and biodistribution (Fig. 5c, e, f). GLA-Bi23SA with homogeneous bi-antennary N-glycans capped with α2-3SA (Fig. 5a) exhibited a markedly extended (3-fold) circulation time (half-life 27.5 ± 0.8 min) (Fig. 5c) and lower enzyme activity in the liver, spleen, and kidney, but the highest level of enzyme in the heart among all

glycoforms tested (90% increase) (Fig. 5e). Importantly, the GLA-LoM6P showed the same trend as would be predicted. The most frequent cause of death in patients with Fabry disease is cardiomyopathy, and increased delivery to the cardiovascular system with glycoforms such as GLA-Bi23SA may present a promising solution. The circulation time of GLA is partly affected by low stability of the enzyme in the plasma at neutral pH. A number of contemporary studies have reported that GLA has

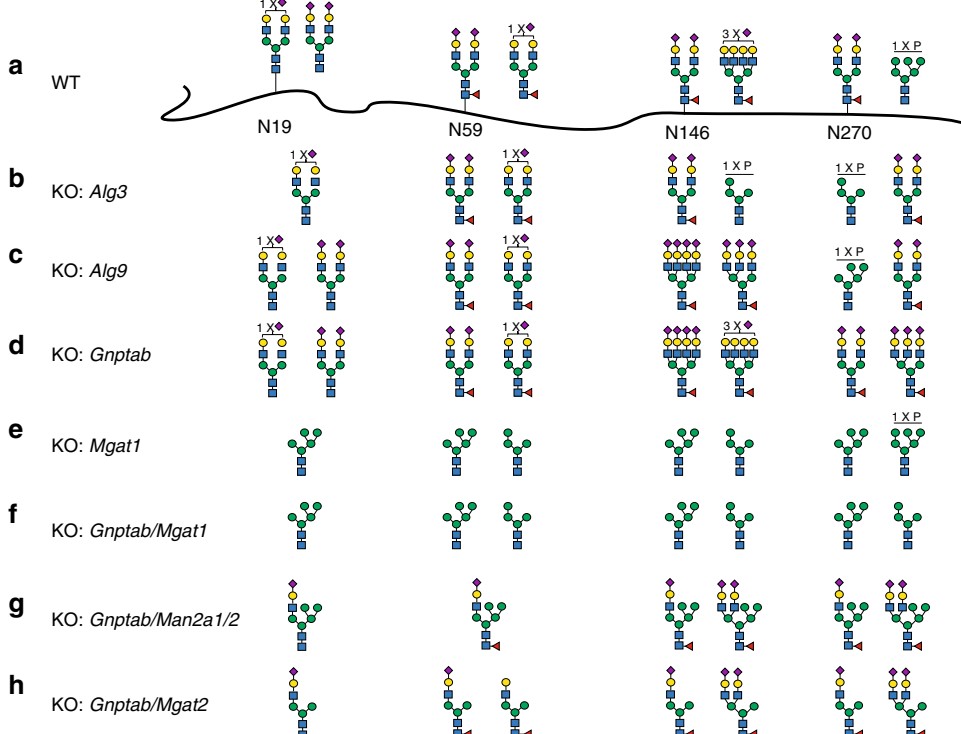

**Fig. 4** Site-specific glycan analyses of β-glucoscerebrosidase (GBA) glycoforms produced in knockout (KO) CHO cells. **a** The two most abundant glycan structures at N-glycosites (N19, N59, N146, and N270) of GBA produced in CHO wild type (WT) are shown, and **b–h** the two most abundant glycans for GBA produced in engineered CHO clones are shown as indicated. Each glycan structure was confirmed by targeted tandem mass spectrometry (MS/MS) analysis. Details regarding the stacking ancestry, sequence analysis, and N-glycans profiling are shown in Supplementary Tables 3 and 4, and Supplementary Fig. 4, respectively

poor stability in unbuffered human plasma in vitro with a loss of more than 50% activity within 15 min[16,42]; however, originally Desnick et al.[25] demonstrated better stability of GLA in neutral-buffered human plasma with a half-life over 200 min. We therefore tested the stability of all GLA glycovariants in mouse plasma with and without buffering with HEPES, and confirmed extended stability with 50% loss of activity at 3 h for Fabrazyme and all GLA variants with buffering (Supplementary Fig. 6). To further analyze the in vivo clearance of GLA variants, we tested mouse plasma by Western blotting (Supplementary Fig. 7), and found that loss of an immunoreactive GLA band migrating at 52 kDa correlated well with loss of enzyme activity and especially slower clearance of GLA-Bi23SA (Fig. 5c).

GLA-26SA with N-glycans capped by α2-6SA demonstrated only marginally elevated circulation time (Fig. 5c), and perhaps surprisingly[43], resulted in higher liver uptake and corresponding decrease in the spleen and kidney uptake (Fig. 5e). The striking increase in liver uptake resembles previous results obtained with albumin neoglyconjugates suggesting interaction of NeuAcα2-6Galβ1-4GlcNAc terminating glycans with the AMR[44], although other studies suggest that primarily NeuAcα2-6GalNAcβ1-4GlcNAc-terminated and -non-sialylated neoglycoproteins are removed from circulation[43]. AMR-mediated uptake of α2-6SA-capped ERTs is predicted to be considerable less efficient compared to the asialo-glycoform based on previous studies demonstrating circulatory half-life of about 1 min for desialy-lated glucocerebrosidase compared to 21 min for the native enzyme[45].

Cellular localization of Fabrazyme and the glycovariants in the heart, kidney, and liver was assessed by immunohistochemistry (IHC) (Fig. 5f). The localization pattern of Fabrazyme in these organs was consistent with that of agalsidase alfa reported in previous studies[17,46]. In the heart, Fabrazyme and all five glycovariants were detected in vascular and/or perivascular cells, but not in cardiomyocytes (Fig. 5f). There were no clear differences between the tested variants. In the kidney, Fabrazyme and GLA-LoM6P, GLA-HiM6P, GLA-HybM6P, and GLA-Bi23SA were predominantly detected in tubular epithelial cells. However, GLA-26SA had significantly decreased number and intensity of positive signals in tubules compared to the other variants tested (Fig. 5f). In the liver, Fabrazyme, GLA-LoM6P, GLA-HiM6P, and GLA-HybM6P were detected in hepatocytes, putative Kupffer cells, and endothelial cells of sinusoidal capillaries. GLA-Bi23SA was also detected in these cell types; however, the number of positive signals in hepatocytes was clearly decreased compared to Fabrazyme. Distribution of GLA-26SA in the liver was remarkably different from other variants; this variant was detected almost exclusively in hepatocytes, and the number of positive signals in hepatocytes was clearly increased compared to Fabrazyme (Fig. 5f).

Encouraged by the improved properties of GLA-Bi23SA with extended circulatory half-life, and GLA-26SA with altered biodistribution, we proceeded to test reduction of accumulated globotriosylceramide (Gb3) substrate in organs 2 weeks after a single injection of 1 mg/kg. We performed two independent series with a single 1 mg/kg dose (Exp. #1) of Fabrazyme and GLA-Bi23SA, and subsequently GLA-26SA (Fig. 5g). GLA-Bi23SA produced the same reduction of the Gb3 content in the heart, kidney, and liver as compared to Fabrazyme (Fig. 5g), and this corresponds to the greatest reduction reported so far with any enzyme strategy used in the Fabry mouse model[16,17,47]. GLA-26SA produced lower reduction of Gb3 levels in the heart and kidney compared to Fabrazyme and GLA-Bi23SA, while the effect in the liver was similar for all variants. This finding correlates

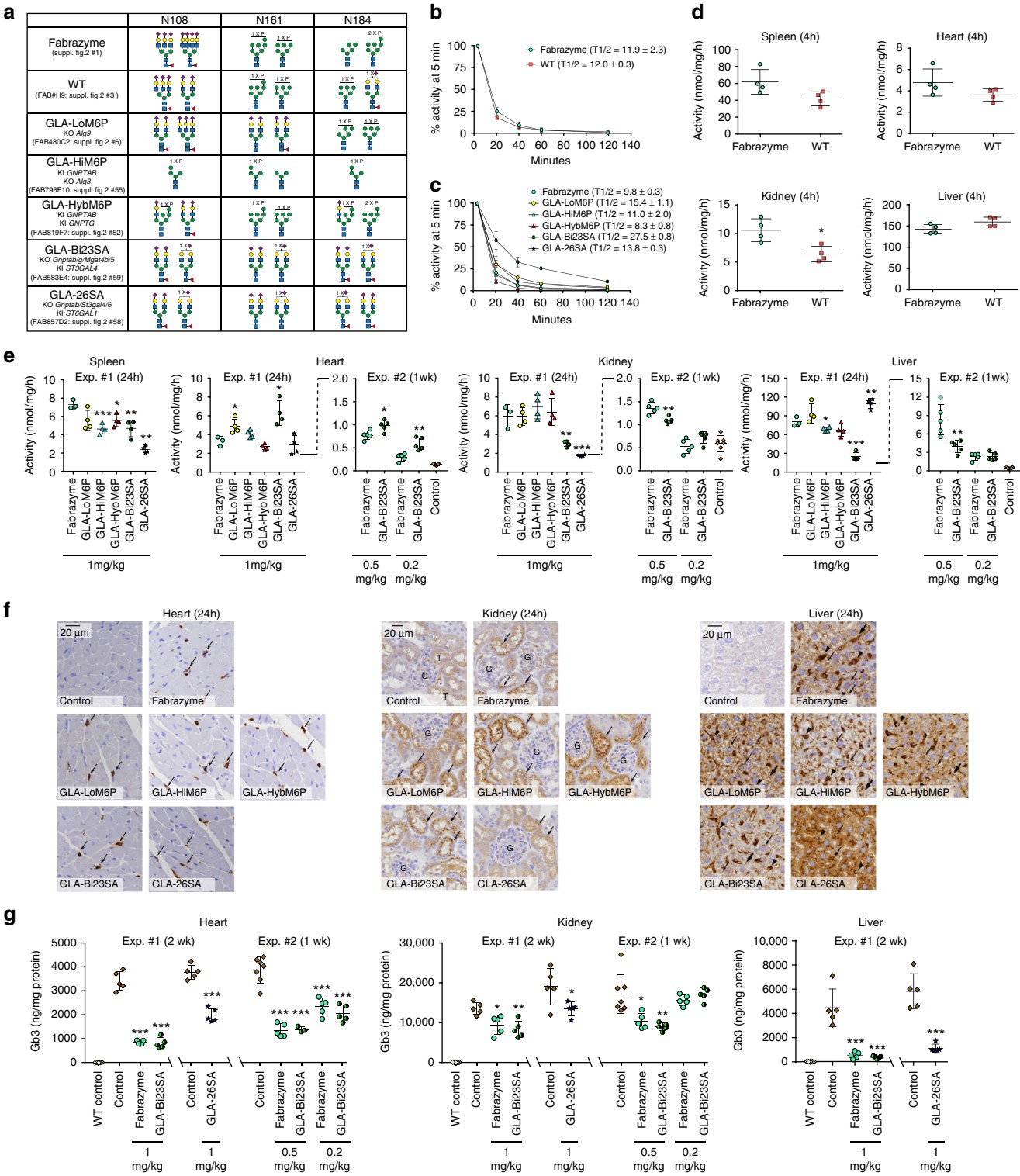

with the lower level of GLA-26SA distributed to the heart and especially kidney (Fig. 5e). We did not observe complete reduction of Gb3 levels in the kidney and heart in agreement with previous reports[17,47], and the reason for the incomplete reduction is unclear and may be due to multiple factors, including that Gb3 and the Globo-series glycolipids are particularly abundant in the kidney[17,47]. Importantly, previous studies using higher dose or repeated infusion of Fabrazyme or other GLA variants have not resulted in further complete reduction of Gb3 in the heart and especially in the kidney[47].

To further explore the performance of GLA-Bi23SA, we tested two lower doses (0.5 and 0.2 mg/kg) in comparison with Fabrazyme (Expt. #2), where we analyzed enzyme activity in organs after 1 week instead of 24 h (Fig. 5e) and residual Gb3 after 1 week instead of 2 weeks (Fig. 5g). This confirmed the improved distribution of GLA-Bi23SA to the heart and lower uptake in the liver compared to Fabrazyme, but also demonstrated dose dependency, suggesting that further studies of optimal dosing for GLA-Bi23SA is needed. More importantly, this clearly demonstrated detectable levels of enzyme activity

**Fig. 5** In vivo study of different α-galactosidase A (GLA) glycovariants in Fabry mice. **a** Summary of the glycan features of GLA glycovariants used. Detailed structures shown in Supplementary Fig. 2 (#1, 3, 6, 55, 52, 59, 58). **b** Fabrazyme and GLA produced in our wild-type (WT) CHO cell line expressed as % of activity at 5 min after injection ($n = 4$). **c** Time-course analysis of GLA activities in plasma after infusion of engineered GLA variants ($n = 4$ except for Fabrazyme in **c** where $n = 3$). **d** GLA enzyme activity in indicated organs 4 h after infusion of Fabrazyme and GLA produced in our WT CHO cell line using the same mice after analysis of plasma activities as shown in **b**. **e** GLA enzyme activity in organs after infusion of engineered different GLA variants as indicated. In one series of mice (Exp. #1), activity was determined 24 h after a single infusion of 1 mg/kg ($n = 5$). In another series of mice (Exp. #2) activity was determined 1 week after a single infusion of 0.5 or 0.2 mg/kg ($n = 5$). Note the change in y-axis scale for the second series. Control in Expt. #2 refers to Fabry mice infused with saline. **f** Immunohistochemistry (IHC) analysis with polyclonal anti-GLA antibody. Annotations used: liver—hepatocytes (small arrows), putative Kupffer cells (arrowheads), endothelial cells of sinusoidal capillaries (large arrows), and punctate lysosome-like distribution of positive signals (small arrows); kidney—cortical tubules (indicated as "T"), glomeruli (indicated as "G"), and tubular epithelial cells (arrows); heart—vascular and perivascular cells (arrows). Scale bars are 20 μm. **g** Globotriosylceramide (Gb3) substrate levels in organs quantified by mass spectrometry from mice treated with Fabrazyme and GLA variants with single doses as indicated. Note that different groups of mice were used for GLA-Bi23SA and GLA-26SA as indicated. WT control refers to wild-type mice, and Control to Fabry mice infused with saline. Error bars are presented as mean with standard deviation. Statistical analysis was performed with the Student's t test against Fabrazyme. *$P < 0.05$, **$P < 0.01$, ***$P < 0.001$. Source data are provided as a Source Data file

1 week after infusion and thus providing strong evidence in support of lysosomal delivery of GLA-Bi23SA in agreement with the immunohistochemistry results (Fig. 5f). The analysis of Gb3 levels in the heart and kidney confirmed the efficient function of both GLA-Bi23SA and Fabrazyme, but also showed that further studies of dosing are needed. These results unequivocally demonstrate that glycoforms of GLA without the classical receptor ligands M6P and Man are efficiently taken up by cells, delivered to the lysosome, and function in reduction of Gb3.

In summary, the glycoengineered GLA variants exhibited distinct pharmacodynamic profiles in Fabry mice. The α2-3SA sialylated design without M6P and terminal Man led to reduced uptake in the liver, prolonged plasma half-life, and improved delivery to the heart. In contrast, the α2-6SA sialylated design led to preferential delivery to hepatocytes and decreased uptake by renal tubular cells. The longer circulation time of GLA-Bi23SA is likely to provide opportunity for wider organ distribution as evidenced by the marked increase in uptake in the heart, and may also provide opportunity for lower dose or less frequent dosing of replacement enzymes, although further studies are needed to address this.

## Discussion

The comprehensive engineering performed with GLA and GBA in CHO cells demonstrates ample options for fine-tuning all key features of N-glycans on lysosomal enzymes known to be important for their cellular uptake, circulation time, and biodistribution. This includes a high degree of site-specific fine-tuning of M6P stoichiometry, exposure of Man, Gal, and GlcNAc residues, and capping by SA. We provide novel designs for recombinant lysosomal enzymes that lack recognition markers for classical MPRs and MRs, but contain homogenous N-glycans capped by SA. Among these we discovered that the GLA-Bi23SA design offers increased circulation time, efficient cellular uptake, and improved organ distribution in a Fabry mouse model despite the lack of M6P and exposed Man residues. Although these preliminary studies did not demonstrate improved substrate reduction in organs, the results suggest that the α2-3SA design potentially may be used to overcome one of the arguably major obstacle for many ERTs, that is, their rapid clearance from circulation by the liver and spleen. Extended circulation is predicted to enable wider biodistribution and possibly transport across the blood–brain barrier[48]. The achieved control of N-glycosylation in CHO cells meets or surpass the glycoengineering opportunities previously presented with non-mammalian cells and post-production modification strategies[14–19,22–24,40]. The clinical features of LSDs and the organs affected differ greatly as do the biostructural properties of the respective deficient enzymes, and

the design matrix and glycoengineered CHO cells developed here will be valuable tools for production and testing of optimal designs for individual ERTs, in order to improve a class of essential drugs with high costs and poor performance.

CHO cells are the preferred mammalian expression hosts for human therapeutics. Given the recent options for targeted and stable gene engineering of glycosylation capacities in mammalian cells[29], we undertook to explore the glycoengineering options for M6P-modified lysosomal replacement enzymes that represent one of the most complex challenges for the biopharmaceutical industry[3,49,50]. Using GLA as an illustrative example, we dissected virtually all steps in the genetic and biosynthetic control of N-glycosylation and M6P tagging, and found surprising plasticity and control for fine-tuning complex N-glycan patterns even with a degree of glycosite specificity (Figs. 2 and 3). Thus, M6P tagging could be tuned up and down and directed to one (N184), two, or all three N-glycosites of GLA, and importantly combined with different degrees of high-Man or complex sialylated N-glycans. Moreover, we also demonstrated production of glycoforms with homogenous SA capping, but lacking M6P or exposed Man residues.

It has long been clear that the structure of N-glycans on replacement enzymes affects cellular uptake and circulation time by interacting with different cell-surface receptors[51], and that altering the glycan composition can be used to direct organ targeting. This was demonstrated first with targeting of GBA with high-Man structures for the MR on macrophages[52], and ERTs with glycans optimized for targeting specific receptors are already successfully used in the clinic[18,37]. Different strategies have been applied to optimize N-glycans for specific cell and organ targeting requirements. To achieve N-glycans with high degree of Man exposure for MR-mediated liver targeting, for example, for GBA treatment of Gaucher patients, the industry has used plant cells[40], human fibrosarcoma cells combined with N-glycan mannosidase inhibitors[38], and CHO cells combined with postproduction treatment with multiple exoglycosidases[53]. We present engineered CHO cells capable of producing this high-Man glycoform of GBA (Fig. 4f), and importantly also related designs with different degrees of Man exposure and SA capping expected to influence kinetics of uptake and circulation half-life (Fig. 4g). To increase the M6P content in particular for targeting muscle cells, yeast has been used to produce the lysosomal α-glucosidase deficient in Pompe disease[15]. Yeast modify human lysosomal enzymes with Man-Pi-6-Man, but the elegant introduction of an uncovering α-mannosidase enzyme results in the production of α-glucosidase rich in M6P[15]. Other strategies to increase M6P content include in vitro chemical conjugation[22,24], or co-expression of a truncated GlcNAc-1-phosphotransferase α/β

precursor[54]. These strategies do not enable fine control over the content (or site specificity) of M6P and other glycan features including SA capping, and the presented engineering of high-M6P glycoforms in CHO cells fully match these strategies (Fig. 3f, g, h). Other postproduction modification strategies including oxidative reduction of glycans[26] and PEGylation[16] have been applied to reduce glycan-mediated receptor uptake and/or enhance circulation, and these may be met by the presented glycoform designs with homogeneous SA capping, but lacking M6P or exposed Man residues (Fig. 3j). Thus, our study suggests that any of the more complex processes used for production of enzymes required for ERTs in the clinic today or in development[14–19,22–24,40] can be produced simpler and more effective in glycoengineered CHO cells. Moreover, there may be advantages in combining distinct glycoforms of lysosomal enzymes with emerging glycosylation-independent targeting strategies developed for blood–brain barrier transport[37,55,56].

Our understanding of and ability to predict the outcome of interactions between glycoproteins with heterogeneous N-glycans presenting these features and the multiple receptors involved is limited. Numerous studies have explored the binding and uptake of extreme glycoforms, such as high-Man- and high/low-M6P-containing lysosomal enzymes[6–9], but systematic studies investigating the complex interplay between different glycan features have not been possible due to the lack of methods to produce such glycoforms. Studies with, for example, the lysosomal α-mannosidase that contains multiple N-glycans with very low M6P content and exposure of Man when produced in WT CHO cells suggest that limited interaction with the MPRs and MR may be advantageous for wider biodistribution and crossing into the brain, possibly due to extended circulation time[48,57]. Similar findings were observed with postproduction modified enzymes with partially destroyed glycans[26]. Here, we explored five distinct glycoforms of GLA, including two lacking M6P or exposed Man residues in a Fabry disease mouse model, and found significant changes in circulation half-life and biodistribution (Fig. 5). Most significantly, the GLA glycoform with α2-3SA-capped N-glycans not only showed enhanced circulation time but also demonstrated efficient uptake and function in all tested organs with improved distribution to the hard-to-reach heart compared to the leading Fabrazyme variant (Fig. 5e, f). Evaluating the relative organ distributions of glycovariants among the four major visceral organs tested illustrate the substantial improved distribution of GLA-Bi23SA to the heart and other organs except the liver (Supplementary Fig. 8). The mechanism for uptake of the α2-3SA-capped GLA glycoform is not clear, but studies have shown that lysosomal targeting of GLA is not exclusively dependent on M6P tagging[12,58], and endocytic receptors including sortilin (SORT1) and megalin (LRP2) that do not bind glycan features have been shown to serve in the uptake of GLA[59,60]. The 3-fold increase in circulatory half-life for α2-3SA-capped GLA is lower than the increase observed with, for example, oxidative degradation and reduction of the β-glucuronidase[26], but this likely reflects the lower stability of GLA in the plasma[16,25,42]. It may be interesting to explore combining this glycoform with the stabilizing molecular chaperone AT1001[61,62] or pegylation[16], and also to consider therapeutic modalities comprising of multiple distinct glycoforms.

In summary, the comprehensive CHO glycoengineering performed and the design matrix generated for lysosomal enzymes opens for systematic studies on options for improving ERTs by designed glycan features. Past studies have demonstrated the value of changing the structures of glycans on enzymes needed for ERTs, but the full potential has clearly not been met by use of yeast and plant production platforms or postproduction modification strategies. The CHO production platform offer new design capabilities, and the remarkable performance found for GLA capped with SA may represent a new design paradigm for many ERTs.

## Methods

**Establishment of stable CHO clones expressing recombinant human GLA and GBA enzymes**. An expression construct containing the entire coding sequence of human GLA was synthesized by Genewiz, USA. Full-length cDNA of human *GBA* was purchased from Sino Biological Inc., China. Both constructs were subcloned into modified pCGS3 (Merck/Sigma-Aldrich) for glutamine selection in CHOZN GS−/− cells (Sigma). CHO cells were maintained as suspension cultures in serum-free media (EX-CELL CHO CD Fusion, cat. no. 14365C), supplemented with 4 mM L-glutamine in 50 mL TPP TubeSpin® Bioreactors with 180 rpm shaking speed at 37 °C and 5% $CO_2$. Cells were seeded at $0.5 \times 10^6$ cells/mL in T25 flask (NUNC, Denmark) one day prior to transfection. Electroporation was conducted with $2 \times 10^6$ cells and 8 μg endotoxin-free plasmids using Amaxa Kit V and program U24 with Amaxa Nucleofector 2B (Lonza, Switzerland). Electroporated cells were subsequently plated in the 6-well plate with 3 mL growth media, and after 72 h, cells were plated in the 96-well plate at 1000 cells/well in 200 μL Minipool Plating Medium containing 80% EX-CELL® CHO Cloning Medium (cat. no. C6366) and EX-CELL CHO CD Fusion serum-free media without glutamine. High expressing clones were selected by assaying the medium for enzyme activity (GLA) or with an enzyme-linked immunosorbent assay using anti-HIS antibodies (for GBA), and selected clones were scaled-up in serum-free media without L-glutamine in 50 mL TPP TubeSpin® shaking Bioreactors (180 rpm, 37 °C and 5% $CO_2$) for enzyme production.

**Purification of GLA and GBA**. For GLA spent, the culture medium was centrifuged at $500 \times g$ for 20 min, filtered (0.45 μm), diluted 3-fold with 25 mM MES (pH 6.0), and loaded onto a DEAE-Sepharose Fast Flow column (Sigma). The column was washed with 10 column volume (CV) washing buffer (25 mM MES with 50 mM NaCl, pH 6.0) and eluted with 5 CV elution buffer (25 mM MES with 200 mM NaCl, pH 6.0). For mouse studies. GLA was further purified by Mono-Q chromatography. For the HIS-tagged GBA culture, the medium was centrifuged, filtered, and mixed 3:1 (v/v) with 4× binding buffer (200 mM Tris, pH 8.0, 1.2 M NaCl), applied to 0.3 mL packed NiNTA agarose (Invitrogen), and then pre-equilibrated in the binding buffer (50 mM Tris, pH 8.0, 300 mM NaCl). The column was washed with the binding buffer and then eluted with 250 mM imidazole in the binding buffer. Purity and quantification was evaluated by sodium dodecyl sulfate-polyacrylamide gel electrophoresis (SDS-PAGE) Coomassie staining.

**CRISPR/Cas9-targeted KO in CHO cells**. We designed and tested three to four guide RNAs (gRNAs) per gene with a high-throughput workflow[63]. Green fluorescent protein (GFP)-tagged Cas9 nuclease was used to enrich for high Cas9 expression by fluorescence-activated cell sorting (FACS), and the cutting efficiency and indel profile of each gRNA was characterized by Indel Detection by Amplicon Analysis (IDAA)[64]. We developed 43 validated gRNA constructs (Supplementary Table 1) and more than 200 CHO cell clones with different gene engineering design (Supplementary Table 2 and Supplementary Data 1). Gene editing was performed in CHO clones stably expressing GLA or GBA. Cells were seeded at $0.5 \times 10^6$ cells/mL in T25 flask (NUNC, Denmark) one day prior to transfection, and $2 \times 10^6$ cells and 1 μg each of endotoxin-free plasmid DNA of Cas9-GFP and gRNA in the plasmid U6GRNA (Addgene Plasmid #68370) were used for electroporation as described above. Forty eight hours after nucleofection, the 10–15% highest labeled (GFP) pool of cells were enriched by FACS, and after 1 week in culture, cells were single-cell sorted by FACS into the 96-well plate. KO clones were identified by IDAA as described[64], as well as when possible by immunocytology with appropriate lectins or monoclonal antibodies. Selected clones were further verified by Sanger sequencing. The strategy enabled fast screening and selection of KO clones with frameshift mutations, and on average we selected two to five clones from each targeting event. In general, the gene targeting did not substantially affect viability, growth, or productivity in the mutant cell clones. The full list of CRISPR gRNA design and PCR primers used are shown in Supplementary Table 1.

**ZFN/CRISPR-mediated KI in CHO cells**. We used targeted KI with zinc-finger nucleases (ZFNs) (modified ObLiGaRe[65] strategy) or CRISPR/Cas9-facilitated non-homologous end-joining[66] into a CHO Safe-Harbor locus[29,67]. Site-specific CHO Safe-Harbor locus KI was based on ObLiGaRe strategy and performed with 2 μg of each ZFN (Merck/Sigma-Aldrich) tagged with GFP/Crimson[29], and 5 μg donor plasmid with full coding human genes (*ST3GAL4*, *ST6GAL1*, *GNPTAB*, or *GNPTG*). In brief, the EPB69 donor plasmid contained inverted CHO Safe-Harbor locus ZFN binding sites flanking the CMV promoter-ORF-BGH polyA terminator. Mono-allelic-targeted KI clones with one intact allele were selected by IDAA analysis[64]. To stack a second gene into a Safe-Harbor locus, we first designed gRNA for the CHO Safe-Harbor locus flanking the ZFN binding site, followed by transfection with 1 μg of a donor PCR product of gene to be inserted with 1 μg Cas9-GFP and 1 μg gRNA. In brief, the donor PCR product was generated by using EPB69 donor plasmid as template, which contained the CMV promoter-ORF-BGH

polyA terminator. KI clones were screened by PCR with primers specific for the junction area between the donor plasmid and the Safe-Harbor locus. A primer set flanking the targeted KI locus was used to characterize the allelic insertion status, and when possible, KI clones were also screened by immunocytology with lectins and monoclonal antibodies.

**GLA enzyme activity assay**. GLA enzyme activity was measured with 33 mM (unless otherwise specified) p-nitrophenyl-α-D-galactopyranoside at 37 °C for 30 min at pH 4.6 in 20 mM citrate and 30 mM sodium phosphate, and the reaction was quenched with borate buffer (pH 9.8) and released p-nitrophenol was read at 405 nm. A standard curve was generated by using 2-fold serial diluted p-nitrophenol in the same assay condition to calculate the amount of released product.

**Site-specific N-glycopeptide analysis**. Approximately 10 µg of purified GLA or GBA in 50 mM ammoniumbicarbonate buffer (pH 7.4) was reduced with dithiothreitol (10 mM) at 60 °C for 30 min and alkylated with iodoacetamide (20 mM) for 30 min in dark at room temperature. Chymotrypsin digestion was performed at a 1:25 enzyme–substrate ratio. The proteolytic digest was desalted by custom-made modified StageTip columns containing two layers of C18 and one layer of C8 membrane (3M Empore disks, Sigma-Aldrich)[68]. Samples were eluted with 50% methanol in 0.1% formic acid, and then dried in SpeedVac and re-solubilized in 0.1% formic acid. Liquid chromatography MS/MS analysis was performed with an EASY-nLC 1000 LC system (Thermo Fisher Scientific) interfaced via nanoSpray Flex ion source to an Orbitrap Fusion MS (Thermo Fisher Scientific). Briefly, the nLC was operated in a single analytical column setup using PicoFrit Emitters (New Objectives, 75 µm inner diameter) custom packed with Reprosil-Pure-AQ C18 phase (Dr. Maisch, 1.9-µm particle size, 19–21 cm column length). Each sample was injected onto the column and eluted in a gradient from 2 to 25% of Solvent B for 45 min at 200 nL/min (Solvent A, 100% $H_2O$; Solvent B, 100% acetonitrile; both containing 0.1 % (v/v) formic acid). A precursor MS1 scan (m/z 350–2000) of intact peptides was acquired in the Orbitrap Fusion at the nominal resolution setting of 120,000, followed by Orbitrap HCD-MS2 at the nominal resolution setting of 60,000 of the five most abundant multiply charged precursors in the MS1 spectrum; a minimum MS1 signal threshold of 50,000 was used for triggering data-dependent fragmentation events. Targeted MS/MS analysis was performed by setting up a targeted MS$^n$ (tMS$^n$) Scan Properties pane.

Glycopeptide compositional analysis was performed from m/z features using in-house written SysBioWare software[69]. For m/z feature recognition from full MS scans LFQ Profiler Node of the Proteome discoverer 2.1 (Thermo Fisher Scientific) was used. A list of precursor ions (m/z, charge and retention time) was imported as ASCII data into SysBioWare and compositional assignment within 5 ppm mass tolerance was performed. The main building blocks used for the compositional analysis were: NeuAc, Hex, HexNAc, dHex, and phosphate. The most prominent peptides corresponding to each potential glycosite were added as an additional building block for compositional assignment. The most prominent peptide sequence related to each N-glycosite was determined experimentally by comparing the yield of deamidated peptides before and after PNGase F treatment. The peptide sequence was determined by higher-energy collisional dissociation MS/MS and the abundance level was calculated manually as the peak area from extracted ion chromatogram integrating ion current from at least four isotopes of the precursor ion envelope. A list of potential glycopeptides and glycoforms for each glycosite was generated and the top 10 of the most abundant candidates for each glycosite were selected for targeted MS/MS analysis to confirm the proposed structure. Each targeted MS/MS spectrum was subjected to manual interpretation. The same N-glycan composition may represent isobaric structures, so the listed glycan structure were assisted by and in agreement with the literature data predicting enzyme functions of the targeted genes together with useful information in MS/MS fragments.

**Mouse studies**. Fabry mice (~3.5 months male) and WT controls were used as reported previously[17]. All animal procedures were reviewed and approved by the Institutional Animal Care and Use Committee of Baylor Research Institute. All injections were performed via the tail vein with enzymes diluted in saline to a total volume of 200 µL per mouse, and the overall study design was identical to previous reports of studies with Fabrazyme and other GLA variants in this mouse model[17].

**Pharmacokinetics**. Enzyme preparations were injected at a dose of 1 mg/kg body weight. At indicated time points, blood samples were collected from the tail vein, plasma was separated by centrifugation, and then used for enzyme assay[17]. SDS-PAGE Western blot for plasma GLA was conducted with rabbit polyclonal antibody to human GLA (Sigma, HPA000237, 1:1000 dilution) and horseradish peroxidase (HRP)-conjugated goat anti-rabbit immunoglobulins (Dako, P0448, 1:3000 dilution).

**Biodistribution and tissue kinetics**. Enzyme preparations were injected at a dose of 1 mg/kg body weight. In a separate study, we used doses of 0.5 and 0.2 mg/kg as indicated. At indicated time points, mice were perfused with saline (to remove blood), and heart, kidney, liver, and spleen were dissected. The whole organs were

homogenized in 0.2 % Triton/saline for enzyme assay. Protein concentration was measured using the BCA Protein Assay Kit (Pierce).

**Immunohistochemistry**. Enzyme preparations were injected at a dose of 2 mg/kg body weight. Heart, kidney, and liver were harvested 24 h after enzyme infusion. Untreated Fabry mouse tissues were used as negative controls. Tissues were fixed in formalin, embedded in paraffin, and 5-µm sections were made. IHC was performed by the Histopathology and Tissue Shared Resource in Georgetown University (Washington, DC, USA). In brief, after heat-induced epitope retrieval in citrate buffer, sections were treated with 3% hydrogen peroxide and 10% normal goat serum, and were then incubated with rabbit polyclonal antibody to human GLA (Sigma, cat. no. HPA000237) used at 1:300 dilution. After incubation with HRP-labeled polymer conjugated to goat anti-rabbit IgG (Dako, cat. no. K4003), signals were detected by DAB chromogen, and the sections were counterstained with hematoxylin. Signal specificity was verified with control staining, in which the primary antibody incubation was omitted. We also developed a mouse monoclonal antibody 6G8 to human GLA that was used to verify IHC. Briefly purified GLA was used for immunization of mice to develop hybridomas, and the specificity of 6G8 was evaluated in HEK293 cell with and without KO GLA.

**Analysis of Gb3 in organs**. Enzyme preparations or vehicle alone (saline) were injected into 6-month-old female Fabry mice at doses of 1 mg/kg body weight. Heart, kidney, and liver were harvested 2 weeks after a single injection. Tissue Gb3 levels were analyzed by MS as described[17]. In a separate study, we used doses of 0.5 and 0.2 mg/kg as indicated.

**Reporting summary**. Further information on research design is available in the Nature Research Reporting Summary linked to this article.

## Data availability

All mass spectrometry raw data underlying Figs. 1–4 and Supplementary Figs. 2 and 4 have been deposited to the ProteomeXchange Consortium via the PRIDE partner repository with the dataset identifier PXD013140 [https://www.ebi.ac.uk/pride/archive/projects/PXD013140]. The source data underlying Fig. 5b–e, g and Supplementary Figs. 1, 3a–d, 6a–c, 7, 8 are provided as a Source Data file. Other data that support the findings of this study are available from the corresponding author upon request.

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

## Acknowledgements

We thank Thomas Braulke (Hamburg) for discussions and critical reading of the manuscript. This work was supported by the Lundbeck Foundation, Novo Nordisk Foundation, Kirsten og Freddy Johansen Fonden, A.P. Møller Fonden, Læge Sofus Carl Emil Friis og hustru Olga Doris Friis' Legat, Innovation Fund Denmark, and the Danish National Research Foundation (DNRF107).

## Author contributions

W.T. designed, planned, and performed most of the gene targeting experiments and co-wrote the manuscript. Zi.Y. and S.Y.V. performed the site-specific glycan analyses. S.W., M.A.S., J.V.C., L.S., Y-H.C. and Y.N. contributed to gene targeting experiments. L.H., C.K. and E.P.B. contributed to antibody development. U.M. contributed to the design and analyses of experiments. S.J-T., R.S. and J.-S.S. performed the mouse studies. H.C. and Z.Y. designed, planned, and co-wrote the manuscript.

## Additional information

**Competing interests:** The University of Copenhagen has filed patent applications EP/3455635 and US/62724543 partly on the basis of this work. W.T., Y.N., C.K., U.M., E.P.B., S.Y.V., Z.Y. and H.C. are named inventors on one or both of these applications. GlycoDisplay ApS has license rights to the patent applications. W.T., Z.Y., Y.N., E.P.B., C.K. and H.C. have shares in GlycoDisplay Aps. C.K. is an employee of GlycoDisplay ApS. The remaining authors declare no competing interests.

