## [Peer Review File · Nature Communications]

Reviewers' comments:

Reviewer #1, an expert in glycoform engineering and therapeutics (Remarks to the Author):

To the authors:

Tian et al. present a manuscript in which they generated a number of CHO KO and KI clones to enable production of a comprehensive collection of N-glycoforms of a recombinant protein. In casu, they use this panel of CHO cells to produce different glycoforms of GLA, the lysosomal alpha-galactosidase used for treatment of Fabry disease, one of the more common lysosomal storage diseases. They also show that the same glycoengineering principles apply to GBA (beta-glucocerebrosidase), used in the treatment of Gaucher's disease, another lysosomal storage disease. The authors present a number of experiments that explore the pharmacokinetics and biodistribution of several GLA glycoforms to liver, kidney, spleen and heart and compare the ability of one glycoform with the currently marketed Fabrazyme to clear Gb3 in different tissues in a Fabry mouse model.

The major claims made by the authors are

- 1) They enable production of lysosomal enzymes with custom designed N-glycoforms affecting cellular uptake and circulation in a panel of glycoengineered CHO cells. This panel now enables systematic studies towards improving ERT.
- 2) Different glycoforms result in different pharmacokinetics and biodistribution. The mainly alpha-2,3-sialylated GLA glycoform exhibits longer circulation time and improved targeting to heart.
- 3) The authors also claim that this may lead to a paradigm shift in the design of some enzymes used in the treatment of lysosomal storage diseases: until now, the consensus has been that a higher M6P content leads to a better therapy, due to more efficient targeting to the lysosomes. The authors claim that other glycoforms that enable longer circulation and different/better biodistribution, such as the alpha-2,3-sialylated form, could make for better therapeutics as opposed to mainly M6P glycoforms, which is rapidly cleared and mainly targeted to the liver.

Some general remarks:

- Please describe in figure legends/methods which statistics were used (i.e.: which error bars are presented, which tests to compare groups, etc...)!!!
- Use consistent naming throughout to enhance manuscript readability: for example, in Supp. Fig. 3d, CHO clone names like FAB446A2, ... are used. With which mutation combinations do these correspond? With which glycoforms in figure 5 do they correspond?
- Carefully check for spelling mistakes, typos and sentence constructions. There are quite a few of these throughout the manuscript.
- Some representative MS analyses are shown and while I am convinced that the authors did a good job in analyzing all other glycoproteomics, most of the raw data is not shown. I do not think it is necessary to provide these with the manuscript but I would recommend to at least deposit the raw MS data in a public database such as the PRIDE archive, and refer to it in the manuscript.

Moving to the experiments, the authors have done a very impressive job in generating a panel of CHO cell clones which enable the production of a huge number of glycoforms and combinations of glycoforms at different glycosites. There are, however, also several issues that need to be addressed in my opinion:

- 1) The authors describe several mutations in the LLO biosynthesis pathway (e.g.: Alg3, 9, 12, 6, 8). An important concern here is that some of these mutations may result in glycosite under-occupancy (this is known to be the case in yeast). However, that question was not addressed in the manuscript. Since one of the glycoforms that was further used for the in vivo work is an Alg9 KO, it is certainly relevant to assess glycosylation completeness of at least this glycoform. Furthermore, are there any signs that site occupancy might be incomplete in the other mutant CHO lines? The most straightforward way to check this would be to do intact protein MS or at least show protein gels of purified GLA on which it is clearly visible that the protein used in the in vivo

experiments consist of one discrete band.

2) Have the authors checked for the presence of N-glycolylneuraminic acid on the termini of the glycans? If they are present, it would be interesting to see if proteins containing N-glycolylneuraminic acid terminated (2,3 or 2,6) would behave differently than SA terminated glycoforms. If they N-glycolylneuraminic acid is undetected, this should be mentioned.

3) Although subject to debate, there are indications that alpha-gal might also be present on CHO produced proteins after clonal selection

(<https://www.ncbi.nlm.nih.gov/pmc/articles/PMC4005363/>). It would be interesting to see whether the authors could detect any alpha-gal in any of the glycoengineered CHO clones.

4) In Figure 5a, the authors show the main glycoforms at each site for GLA. Could they mention how homogeneous this is? I.e.: what is the percentage of the total amount of GLA that carries that glycan? Is this close to complete or is the mix used for the in vivo experiments still somewhat heterogeneous, carrying more than just the two presented glycoforms?

5) The authors write on p10 that GLA has extremely poor stability in plasma ('loss of more than 50% activity in ~15 min at 37°C'). However, the data presented in Supp. Fig. 6 seem to be in disagreement with this and show a half life around 45 min, without any mention of it in the main text. Is the 15 min on p10 a typo?? If not, the authors should clarify their point of view here, because this discussion is very much relevant to the following issue: If the half life is around 15 min, the interpretation of Figs 5b and c becomes problematic: since the 'clearance' is measured in terms of enzyme activity, one would expect a confounding of enzyme stability and clearance. The 'clearance' speed seems to be around 15 min T1/2 for most glycoforms. This is the same as the claimed stability T1/2, but shorter than the 45 min T1/2 the authors show in Supp Fig 6. The drop in activity for the Bi23SA is markedly slower (Fig 5c) and the logical conclusion would be that it is governed by loss of enzyme activity rather than actual clearance. I would like to see a thorough discussion of this issue.

6) Even with a half life of 45 mins, the interpretation of Fig 5 b and c is still somewhat problematic. To measure clearance from the circulation, the authors should really be measuring protein levels instead of enzyme activity to make claims about the effect of glycans on clearance (provided the enzyme is not degraded too much this could be done with ELISA). If this is not possible, please at least address this issue in the discussion.

7) Although the authors see differences in organ distribution, this does not seem to translate into any difference in Gb3 clearance in the heart or other organs, and there is certainly still room for improvement (also with Fabrazyme for that matter) when comparing with Gb3 levels in WT mice. The authors claim that their findings might cause a paradigm shift in the field (i.e.: 'you don't need phosphorylated glycans to treat LSD's, in fact, you might be better off with other glycans because they result in better biodistribution'). I find these data not convincing enough to make this claim. In any case, 2,3 sialylated GLA is definitely not working any better than Fabrazyme, despite the better biodistribution to the heart. The current view is that enhancing the mannose phosphate content of ERT enzymes will improve their targeting to the lysosome and thus their clinical effect. To challenge this view, the authors should at least have included a glycoform with enhanced phosphorylation in the experiment shown in Fig 5g and show that it is also not better than Fabrazyme. Is it possible that, although a larger amount of GLA-Bi23SA is reaching the heart, a smaller fraction of it is actually entering into the lysosomes and that is why it doesn't work any better than Fabrazyme? Maybe less fabrazyme reaches the heart, but more of it eventually ends up in the lysosomes through the MPR's because it contains more M6P. I think a crucial experiment that is lacking here is an experiment that directly assesses the lysosomal targeting of the different glycoforms. E.g.: confocal microscopy experiments quantifying the colocalization of the different glycoforms with lysosomal proteins, and checking the fraction that does not reach the lysosomes.

8) Apart from this experiment, it would also be interesting to see a more profound discussion on how glycoproteins lacking phosphorylated glycans get to the lysosomes: are they taken up through a cell surface receptor? Pinocytosis? How is the trafficking happening? It should probably be noted that if it is not happening through receptor-mediated transport, the cellular targeting might be much worse in man than in mice, because the surface/volume ratio of the circulatory system is much lower in man and would thus provide much less opportunities for a drug to be taken up through non-receptor mediated processes such as pinocytosis before being cleared.

9) Moreover, another important control to see whether glycans have a meaningful impact on the clearance, biodistribution and uptake (especially since 2,6 sialylation doesn't seem to behave any different than non-sialylated glycoforms) is to see how non-glycosylated GLA behaves. To obtain this, natively folded GLA could be treated with PNGaseF: for many glycoproteins, this works on natively folded protein when more PNGaseF is added than in the standard protocol.

10) Fig 5d and e, why is the comparison between WT and Fabrazyme and between the other mutants and Fabrazyme split out in two different figures and especially two different time scales? I think the 24h WT data should be added to Fig 5e and taken along in the statistical analysis to make for a fair comparison. Are the differences still significant if the WT protein is added? I think the WT data should also be presented for the organ distributions in figure 5f, again to make for a fair comparison. Along the same line: why are the WT and mutant data split out in Figs 5 b and c? It seems that the differences picked up between most of the mutants and Fabrazyme are in the same order of magnitude as the difference between the two fabrazyme datasets in Fig 5b and c, and thus probably not biologically relevant, except maybe for the GLA Bi23SA.

11) On p11 the authors state that they picked out GLA-Bi23SA for animal experiments, based on its unique behavior. The authors should specify exactly what they mean with this statement, because, in the preceding paragraph (and in Fig 5f) they state and show that it behaves similar to the other glycoforms, except maybe in liver. Personally, I don't see much difference between GLA-Bi23SA and Fabrazyme (maybe the micrographs are too small). On the other hand, the measured activities (Fig 5e) do not always seem to agree with the tissue slices. Here, contrary to the impression I get from the slides, GL-Bi23SA shows significantly higher activity in the heart. In kidney, much lower activity than fabrazyme, but on the micrograph, it looks the same to me. On the other hand, GLA-Bi26SA seems to have a more unique behavior to me, and opposite to GL-Bi23SA. Given the fact that Fabrazyme and GLA-Bi23SA don't show any difference in clearance of Gb3 from heart, it would be interesting to see if e.g. GLA-Bi26SA really results in higher Gb3 levels, or that the glycoform doesn't really matter.

12) Given the very low stability in plasma (half life of 15 or 45 mins?), it is surprising the authors can measure enzyme activity after 24h in the different organs. Could the authors discuss this and also why they chose to measure enzyme activities in the organs and biodistribution after 24h and Gb3 clearance only after 2 weeks? Would any differences not be much more pronounced if the measurements were done shortly after administering the enzymes? From Figs 5 b and c it is clear that most of the enzyme has been cleared from the circulation after 2 hours. One would think the biodistribution would be more informative at such a time interval than after 24 h, when clearance from/breakdown in an organ is maybe also playing an important role? Concerning Gb3 clearance, do the authors have an idea when the Gb3 levels reach a minimum value? Is this minutes, hours, days or weeks after administration?

13) More on protein clearance and uptake; How do protein clearance and uptake through lectins such as MR, AMR, MPR differ in man and mouse? I could imagine there might be differences, considering that normal mice proteins can have different terminating residues (alpha-gal, differences in sialic acid linkage, N-glycolyl neuraminic acid?). In other words, could the authors discuss how they think their findings in the mouse model would translate to human patients?

Summarizing: The authors have done a considerable effort in generating and analyzing a panel of CHO clones to enable the production of a host of different glycoforms. They have produced GLA in some selected clones and assessed the pharmacokinetics, biodistribution and Gb3 clearance of several glycoforms in mice. I think there are interesting results here that are potentially very important and could maybe cause a paradigm shift in the field. In particular, the results showing similar Gb3 clearance for Fabrazyme and GLA-Bi23SA in Fabry mice are intriguing. It is a pity that the evidence is insufficient and thus not entirely convincing to show that M6P is not necessarily the best choice for ERT enzymes. I think these issues can be addressed by a more in depth discussion of several elements and some additional experiments.

Reviewer #2, an expert in metabolic glycoengineering (Remarks to the Author):

The authors reported an excellent and thorough work that expanded the design space for engineering lysosomal replacement enzymes. According to the authors, their report is the first evidence that glycoforms capped with α 2-3 linked sialic acid but not α 2-6 sialic acid exhibit improved circulation in addition to other pharmacodynamic properties (page 4). However, an earlier report (Unverzagt et al., *J. Med. Chem.* 2002, 45, 478) provided evidence that α 2-3 sialylated neoglycoproteins have longer serum half-life than their α 2-6 analogues. The authors should discuss their results within the context of Unverzagt and coworkers' earlier work.

Speculating on the mechanism of uptake of α 2-3SA capped glycoforms, the authors cited Markmann et al. work (reference 17) that lysosomal targeting of GLA is at least partly independent on M6P-tagging. The clause "at least partly" is confusing because Markmann et al. explicitly conclude that the targeting is M6P-dependent, and they did not restrict their findings to α 2-3SA capped glycoforms. The authors should address this.

The authors failed to provide the rationale for selecting the five glycoforms tested in the in vivo experiments. Expanding the library of the tested glycoforms may provide a cue to the superior performance of the α 2-3SA, particularly the glycoforms without core fucose (Fig 3k, l and supplementary Fig 2 panel 60 and 61). Testing the non-fucosylated glycoforms may provide answers to the lower serum half-life of α 2-6SA compared with α 2-3SA given the reports of Unverzagt et al. (*J. Med. Chem.* 2002, 45, 478) that core fucosylation of a biantennary α 2,6-sialylated N-glycan significantly accelerates neoglycoprotein clearance from the bloodstream.

It will be interesting to address the microheterogeneity observed in α 2-6SA glycoform (supplementary Fig 2, panel 58) and how this can affect clearance from the bloodstream (please see *J. Med. Chem.* 2002, 45, 478).

Some studies show the superior pharmacokinetic performance of the α 2-6SA glycoform relative to the α 2-3SA (please see the review, Tejwani et al. *Biotechnol. J.* 2018, 13, 1700234, and the relevant reference therein). It will, therefore, be informative to test the effect of this α 2-6SA glycoform on the reduction of accumulated globotriaosylceramide (page 11) as they did for the α 2-3SA analogue. The results may substantiate their conclusion that uptake is independent of M6P and Man receptors as they alleged.

Minor points:

Page 3: "N-Acetyl-glucosamine (GlcNAc) and N-Acetyl-galactosamine (GalNAc) are typical written as "N-acetylglucosamine" and "N-acetylgalactosamine"

Page 10 (and maybe elsewhere): "mgs/L" should be "mg/L"

Page 15 (and elsewhere): the standard abbreviation for hours is "h" (not hrs)

Page 15, last paragraph: should there be a comma after "For GLA" at the start (also in this paragraph "ml" is used ("mL" is mostly used elsewhere, please be consistent)

Page 17 – heading of "Data Analysis" – based on usual formatting this would be "Data analysis"

References – check for consistent formatting (e.g., References 6, 17, 29 [and others] use upper case throughout the article title)

Re: NCOMMS-18-25490-T

The Glycosylation Design Space for Recombinant Lysosomal Replacement Enzymes Produced in CHO Cells

Point-by-point response & action list to reviewers comments

Reviewer #1

General remarks #1: Please describe in figure legends/methods which statistics were used (i.e.: which error bars are presented, which tests to compare groups, etc...)!!!

Response #1: Thank you.

Action #1: The statistics information is now included in legend to Fig. 5 as follows:

“Statistical analysis was performed with the Student’s t-test between groups. Error bars are presented as mean with standard deviation.”

General remarks #2: Use consistent naming throughout to enhance manuscript readability: for example, in Supp. Fig. 3d, CHO clone names like FAB446A2, ... are used. With which mutation combinations do these correspond? With which glycoforms in figure 5 do they correspond?

Response #2: Agree.

Action #2: All references to CHO clone names including Supp. Fig.3e now include KO/KI gene designs.

General remarks #3: Carefully check for spelling mistakes, typos and sentence constructions. There a quite a few of these throughout the manuscript.

Response #3: Thank you.

Action #3: Fixed to our best ability.

General remarks #4: Some representative MS analyses are shown and while I am convinced that the authors did a good job in analyzing all other glycoproteomics, most of the raw data is not shown. I do not think it is necessary to provide these with the manuscript but I would recommend to at least deposit the raw MS data in a public database such as the PRIDE archive, and refer to it in the manuscript.

Response #4: Agree.

Action #4: All the raw MS data will be uploaded to the PRIDE archive upon acceptance. Following statement included p20:

“Data availability

All mass spectrometry raw data are available on PRIDE PDXXXX (to be inserted upon acceptance).”

Query #1: The authors describe several mutations in the LLO biosynthesis pathway (e.g.: Alg3, 9, 12, 6, 8). An important concern here is that some of these mutations may result in glycosite under-occupancy (this is known to be the case in yeast). However, that question was not addressed in the manuscript. Since one of the glycoforms that was further used for the in vivo work is an Alg9 KO, it is certainly relevant to assess glycosylation completeness of at least this glycoform. Furthermore, are there any signs that site occupancy might be incomplete in the other mutant CHO lines? The most straightforward way to check this would be to do intact protein MS or at least show protein gels of purified GLA on which it is clearly visible that the protein used in the in vivo experiments consist of one discrete band.

Response #1: We agree. We did observe low occupancy at the N184 site (GLA) with inactivation of *Alg5* and *Alg6* by the site-specific (glyco)peptide MS analysis, but not at the N108 and N161 sites. However, we did not observe significant reduction in occupancy with inactivation of *Alg3* or *Alg9*.

Action #1: The following sentences have been included in the Results section (p.5 third paragraph) to clarify analyses of occupancy:

“The occupancy at individual N-glycosites was evaluated by comparing the peptide/glycopeptide ratio for each N-glycosite.”

“In general the occupancies at the three glycosites were found to be near complete for GLA produced in the CHO mutants with the exception of the N184 glycosite where KO of *Alg5* and *Alg6* reduced the occupancy.”

We have also included SDS-PAGE Coomassie analysis of the five GLA glycoforms used for the animal studies in Supplementary Fig. 6a illustrating homogeneous bands.

Query #2: Have the authors checked for the presence of N-glycolylneuraminic acid on the termini of the glycans? If they are present, it would be interesting to see if proteins containing N-glycolylneuraminic acid terminated (2,3 or 2,6) would behave differently than SA terminated glycoforms. If they N-glycolylneuraminic acid is undetected, this should be mentioned.

Response #2: We have not checked for NeuGc systematically as this is an old problem associated with use of bovine serum in culture medium, and our XIC screening for glycan fragment ions showed that the most common contributions were from HexNAc (204/186/168) and NeuAc (292/274). The CHO clones used in this study were grown exclusively in serum-free media.

Action #2: None.

Query #3: Although subject to debate, there are indications that alpha-gal might also be present on CHO produced proteins after clonal selection (<https://www.ncbi.nlm.nih.gov/pmc/articles/PMC4005363/>). It would be interesting to see whether the authors could detect any alpha-gal in any of the glycoengineered CHO clones.

Response #3: We have not checked for α Gal systematically as this is quite an elaborate task to exclude the reported putative 0.2%. The study by Ganesh Venkataraman and colleagues is quite controversial and essentially negated by the decades safe use of many different CHO clones in production of glycoprotein therapeutics. We have chosen not to discuss this in order to prevent further propagation of this story, and this is also partly because the experimental detailed conditions under which the study was performed was not disclosed in the published Letter to the Editor by Venkataraman. The RNAseq data generated on a limited number of mutant clones we have obtained does not indicate expression of the *Ggta* α 3Gal-T gene, and clearly the MS analysis data obtained demonstrate that if present any α Gal capping must be minimal. It is possible in principle that selection of a CHO clone with low levels of expression of the *Ggta* gene or induction of the *Ggta* gene during gene engineering could result in minor amounts of α Gal capping. However, we do not think this is worth to consider in the present systematic screening study of glycoengineering as this would not materially affect the outcome and conclusions drawn. If/when a particular glycoengineering design were to be used for production of clinical enzyme detailed analysis would be needed, and in case of the remote possibility that minor amounts of α Gal, NeuGc (See query/response #2), or other unexpected glycan features are detected it is simple to eliminate the corresponding genes.

Action #3. None

Query #4: In Figure 5a, the authors show the main glycoforms at each site for GLA. Could they mention how homogeneous this is? I.e.: what is the percentage of the total amount of GLA that carries that glycan? Is

this close to complete or is the mix used for the *in vivo* experiments still somewhat heterogeneous, carrying more than just the two presented glycoforms?

Response #4: The detailed N-glycan profiling data with percentages of the 5 major glycoforms are shown in Supplementary Fig. 2, Panels 1, 3, 6, 55, 52, 59 and 58, which is indicated in Figure 5a.

Action #4: The reference to the Supplementary data has been emphasized in the text and Fig. legends.

Query #5: The authors write on p10 that GLA has extremely poor stability in plasma ('loss of more than 50% activity in ~15 min at 37°C). However, the data presented in Suppl. Fig. 6 seem to be in disagreement with this and show a half-life around 45 min, without any mention of it in the main text. Is the 15 min on p10 a typo?? If not, the authors should clarify their point of view here, because this discussion is very much relevant to the following issue: If the half life is around 15 min, the interpretation of Figs 5b and c becomes problematic: since the 'clearance' is measured in terms of enzyme activity, one would expect a confounding of enzyme stability and clearance. The 'clearance' speed seems to be around 15 min T1/2 for most glycoforms. This is the same as the claimed stability T1/2, but shorter than the 45 min T1/2 the authors show in Suppl Fig 6. The drop in activity for the Bi23SA is markedly slower (Fig 5c) and the logical conclusion would be that it is governed by loss of enzyme activity rather than actual clearance. I would like to see a thorough discussion of this issue.

Response #5: We agree that this is an important point that needs further elaboration in the text. The sentence in the text p.10 refers to *in vitro* stability data in unbuffered human plasma reported in the two citations (references 22 and 60: Kizhner, T. *et al* 2015; Sakuraba, H. *et al* 2006). The data presented in the original manuscript Suppl. Fig. 6 was conducted with mouse plasma to relate to the mouse studies performed in this study. We do not know why there is an apparent discrepancy between these values, but we have independently become aware of one original seemingly forgotten study (Desnick *et al.* 1979; Current reference 30) demonstrating that stability of GLA in buffered plasma is greater with half-life of > 200 min. We have therefore redone the *in vitro* stability studies with HEPES buffered mouse plasma and now confirmed longer stability with half-life around 3 h for all the tested glycovariants. Thus, this issue is now less relevant, although the stability of GLA in plasma *in vivo* may still converge with clearance.

Action #5: A new Suppl. Fig. 7 with stability of GLA in HEPES buffered mouse plasma is now included, and the following text has been modified (corresponding to the text on p.11 in the original manuscript):

“The circulation time of GLA is partly affected by low stability of the enzyme in plasma at neutral pH. A number of contemporary studies have reported that GLA has poor stability in unbuffered human plasma *in vitro* with loss of more than 50% activity within 15 min^{22,60}, however, originally Desnick and colleagues³⁰ demonstrated better stability of GLA in neutral buffered human plasma with half-life over 200 min. We therefore tested the stability of all GLA glycovariants in mouse plasma with and without buffering with HEPES, and confirmed extended stability of over 3 h for Fabrazyme and all GLA variants in buffered plasma (Suppl. Fig. 6). To further analyze the *in vivo* clearance of GLA variants we tested mouse plasma by SDS-PAGE Western blotting (Suppl. Fig. 7), and found that loss of an immunoreactive GLA band migrating at 52 Kd correlated well with loss of enzyme activity and especially slower clearance of GLA-Bi23SA (Fig. 5c). Interestingly, a pegylated version of GLA (PRX-102) with an approximate 2-fold enhanced stability in non-buffered human plasma has a similar clearance and biodistribution profile in the Fabry mouse model²².”

Query #6: Even with a half life of 45 mins, the interpretation of Fig 5 b and c is still somewhat problematic. To measure clearance from the circulation, the authors should really be measuring protein levels instead of enzyme activity to make claims about the effect of glycans on clearance (provided the enzyme is not degraded too much this could be done with ELISA). If this is not possible, please at least address this issue in the discussion.

Response #6: To our knowledge antibodies suitable for developing a capture ELISA assay to detect intact GLA are not available. We now present SDS-PAGE WB analysis (Suppl. Fig. 7) that demonstrates loss of GLA protein in agreement with the observed loss of activity.

Action #6: We refer to Response & Action to Query #5.

Query #7: Although the authors see differences in organ distribution, this does not seem to translate into any difference in Gb3 clearance in the heart or other organs, and there is certainly still room for improvement (also with Fabrazyme for that matter) when comparing with Gb3 levels in WT mice. The authors claim that their findings might cause a paradigm shift in the field (i.e.: ‘you don’t need phosphorylated glycans to treat LSD’s, in fact, you might be better off with other glycans because they result in better biodistribution’). I find these data not convincing enough to make this claim. In any case, 2,3 sialylated GLA is definitely not working any better than Fabrazyme, despite the better biodistribution to the heart. The current view is that enhancing the mannose phosphate content of ERT enzymes will improve their targeting to the lysosome and thus their clinical effect. To challenge this view, the authors should at least have included a glycoform with enhanced phosphorylation in the experiment shown in Fig 5g and show that it is also not better than Fabrazyme. Is it possible that, although a larger amount of GLA-Bi23SA is reaching the heart, a smaller fraction of it is actually entering into the lysosomes and that is why it doesn’t work any better than Fabrazyme? Maybe less Fabrazyme reaches the heart, but more of it eventually ends up in the lysosomes through the MPR’s because it contains more M6P. I think a crucial experiment that is lacking here is an experiment that directly assesses the lysosomal targeting of the different glycoforms. E.g.: confocal microscopy experiments quantifying the colocalization of the different glycoforms with lysosomal proteins, and checking the fraction that does not reach the lysosomes.

Response #7: We would first like to stress that the primary focus and achievement of the study is the control over production of distinct glycoforms of lysosomal enzymes, which now enables studies of the performance of distinct glycoforms and glycan features for this class of enzymes. The secondary goal is to illustrate an example of the potential therapeutic value of this platform. Our study with GLA as the demonstration model clearly demonstrates that the general belief that M6P is required for uptake is incorrect. GLA with N-glycans homogeneously capped by 2-3SA demonstrates efficient uptake and function in the lysosomes in multiple organs. The study is clearly limited in design and scope given the primary objective, but we unambiguously demonstrate longer circulatory half-life and improved biodistribution with less enzyme being distributed to the liver. This is fully supported by the additional studies now included with lower doses (0.5 and 0.2 mg/kg) and the extended time-point for analysis (1 wk) of enzyme activities in organs. Our statements:

In the Summary: “We demonstrate distinct predicted organ distribution and circulation time of different glycoforms of α -galactosidase A in a Fabry disease mouse model, and find that an α 2-3 sialylated glycoform designed to eliminate uptake by the mannose 6-phosphate and mannose receptors exhibits improved targeting to hard-to-reach organs such as heart, and longer circulation time. This may suggest a paradigm shift in design of some replacement enzymes, and the developed design matrix and engineered CHO cell lines now enables systematic studies towards improving enzyme replacement therapeutics.”; and

In the Discussion: “The CHO production platform offers new design capabilities, and the remarkable performance found for GLA capped with SA may represent a new design paradigm for many ERTs.”;

are in agreement with the data and do suggest a paradigm shift for the design of ERTs. We fully agree that the data in the Fabry mouse model does not allow us to conclude that GLA-Bi23SA is “working better” than Fabrazyme in terms of reduction of Gb3, and we believe we carefully worded our statements to not state this, but the new design certainly hold promise for improving ERTs.

The Reviewer agrees that the current view is that enhancing the M6P content of ERTs will improve lysosomal targeting and clinical effect, but it is important to consider that a large number of studies in the past have addressed higher M6P content by different means with several lysosomal enzymes including GLA (e.g. Tsukimura et al. Molecular Medicine 2012) without being able to demonstrate superior reduction in

accumulated Gb3 in relevant animal models. Importantly, most of these studies demonstrate enhanced uptake and substrate reduction in skin fibroblasts. Thus, we believe that it is fair to discuss a possible paradigm shift to consider in the design of ERTs.

We selected the α 2,3 SA capped glycoform (GLA-Bi23SA) as demonstration model for the animal studies because: i) GLA-Bi23SA clearly exhibited superior circulation time; ii) it has not been possible to produce this glycoform before; and iii) the performance of this glycoform has therefore not previously been studied. The animal studies are of course limited by cost and ethical considerations, so it is not possible to perform studies without a relevant and important hypothesis for potential drug development. The fundamental novel finding from our experiments is that GLA without M6P and exposed Man actually results in efficient cellular uptake *in vivo* with an improved biodistribution defined as significantly lower enzyme distribution to the liver. Moreover, with the dose and time points tested we found the same reduction in residual Gb3 in the organs tested compared to Fabrazyme. Clearly, further studies are needed to demonstrate performance in the clinic, but it is well accepted that the Fabry mouse is a poor model. Most importantly, in all past studies in this mouse Fabry model with different GLA variants and use of different doses and/or repeat dosing, the absolute reduction in residual Gb3 levels in heart and kidney achieved is essentially identical to what we achieved in the present study. The reason for this is currently not understood, but clearly could relate to subpopulations of cells in organs that do not have Gb3 accumulation/uptake the ERT (Bangari *et al.* Am J Pathol 2015). Very recently, a rat model has been suggested to be a better representative disease model (Miller *et al.* FASEB J 2018), and we hope to test this in the future.

To address the Query raised we also conducted an additional series of mouse studies with lower enzyme doses (0.2 and 0.5 mg/kg) of Fabrazyme and GLA-Bi23SA (Fig. 5). In this study we tested the activity of targeted enzymes after 1 wk compared to 24 h in the first series of studies, and the results reproduced the higher level of GLA activity in the heart found with GLA-Bi23SA compared to Fabrazyme. Using two lower doses we found that the reduction in Gb3 levels was similar for Fabrazyme and GLA-Bi23SA, and further that there was dose dependency. These studies clearly demonstrate that GLA-Bi23SA is taken up into the lysosome and functions as well as Fabrazyme at the same doses in the organs tested.

Action #7: We have modified the overall text to stress the above points (with additional references) and included new animal data in Figure 5. Please see multiple yellow highlighted sections.

Query #8: Apart from this experiment, it would also be interesting to see a more profound discussion on how glycoproteins lacking phosphorylated glycans get to the lysosomes: are they taken up through a cell surface receptor? Pinocytosis? How is the trafficking happening? It should probably be noted that if it is not happening through receptor-mediated transport, the cellular targeting might be much worse in man than in mice, because the surface/volume ratio of the circulatory system is much lower in man and would thus provide much less opportunities for a drug to be taken up through non-receptor mediated processes such as pinocytosis before being cleared.

Response #8: We agree. We did discuss this briefly (original Discussion p.14) with references to studies demonstrating that both sortilin and megalin serve as endocytic receptors for GLA (Previously references 70, 71; Current references 74, 75). The query regarding surface/volume ratio of the circulatory system in mouse relates to the relatively higher dose used in man where the current 1 mg/kg converts into approx. 12.3 mg/kg in the mouse.

Action #8: The Discussion paragraph in p.16 has been modified as follows:

“The mechanism for uptake of the α 2-3SA capped GLA glycoform is not clear, but studies have shown that lysosomal targeting of GLA is not exclusively dependent on M6P-tagging^{17, 73}, and endocytic receptors including sortilin (*SORT1*) and megalin (*LRP2*) that do not bind glycan features have been shown to serve in uptake of GLA^{74, 75}.”

Query #9: Moreover, another important control to see whether glycans have a meaningful impact on the clearance, biodistribution and uptake (especially since 2,6 sialylation doesn't seem to behave any different than non-sialylated glycoforms) is to see how non-glycosylated GLA behaves. To obtain this, natively folded GLA could be treated with PNGaseF: for many glycoproteins, this works on natively folded protein when more PNGaseF is added than in the standard protocol.

Response #9: We disagree. While the α 2-6SA glycoform exhibited lower circulation time and enhanced targeting to the liver, we do not provide data that indicates the effects would be the same as GLA with non-sialylated N-glycans. Clearly the trend would be expected to be the same, but the clearance may be much faster. Studies with neoglycoproteins support this latter prediction (Park *et al.* 2005; Reference 58). The suggestion to remove N-glycans entirely is academic and not justifiable, and it would require confirmatory studies that the fold and structure of the treated enzyme was intact.

Action #9: None.

Query #10: Fig 5d and e, why is the comparison between WT and Fabrazyme and between the other mutants and Fabrazyme split out in two different figures and especially two different time scales? I think the 24h WT data should be added to Fig 5e and taken along in the statistical analysis to make for a fair comparison. Are the differences still significant if the WT protein is added? I think the WT data should also be presented for the organ distributions in figure 5f, again to make for a fair comparison. Along the same line: why are the WT and mutant data split out in Figs 5 b and c? It seems that the differences picked up between most of the mutants and Fabrazyme are in the same order of magnitude as the difference between the two fabrazyme datasets in Fig 5b and c, and thus probably not biologically relevant, except maybe for the GLA Bi23SA.

Response #10: The data in Figure 5d and e represent two different experiments and cannot be combined because two different time-points (4 h and 24 h) were used. Figure 5d only reports a direct comparison of Fabrazyme with GLA produced in our CHO cell line. As discussed in the text and shown in Figure 5a (and Suppl. Fig. 2 #1,2) we observed distinct differences in N-glycosylation between Fabrazyme with GLA produced in our CHO cell line, and corresponding slight differences in circulation and organ distribution, although only the difference in kidney enzyme was significant. We chose to baseline all GLA variants against Fabrazyme given the large number of reports in the literature available for comparison. It is important to note that the heterogeneity in glycosylation of Fabrazyme and similar wildtype variants prevents direct structure-function interpretations and conclusions about involved interactions with glycan binding receptors. Moreover, GLA produced in CHO cells has a natural high content of M6P, and GLA is therefore not the best model for studying higher content of M6P, although we could clearly increase the content. As pointed out though the variants with complete lack of M6P and instead capped with α 2-3/2-6SA produced substantially different circulation and organ distribution, and the same trend was found for GLA-LoM6P in the heart. These differences are clearly biologically relevant as also demonstrated by the new additional studies included with lower doses and extended evaluation timepoints (Fig. 5e,g).

Action #10: We have included the following sentence in Results (p.10):

“We first benchmarked GLA produced in our CHO WT cell (~100 mg/L) with a clinical lot of Fabrazyme finding lower content of exposed Man residues on GLA produced in our CHO WT cells (**Fig. 5a and Supplementary Fig. 2, Panels 1, 3**). The GLA enzyme produced in our CHO WT cells exhibited similar blood circulation half-time (12.0 ± 0.3 min) as Fabrazyme (11.9 ± 2.3 min) (**Fig. 5b**), but with trends of higher liver targeting and lower spleen, kidney and heart targeting of our GLA variant compared to Fabrazyme (4h after infusion), although only the lower kidney targeting was significant (**Fig. 5d**). Given the slight differences in glycosylation found between Fabrazyme and GLA produced in our CHO WT cells, we chose to use Fabrazyme for comparison in further studies to enable direct comparison with results from the literature.

We then tested five distinct glycoforms of GLA predicted to provide the broadest range of potential interactions with the major lectin receptors, M6PR, MR, and AMR, known to interact with and affect uptake and circulation of glycoproteins in blood (Fig. 5a).”

And modified/added the following text to the legend of Figure 5 to clarify this:

“(b) Time-course analysis of GLA activities in plasma after infusion of Fabrazyme and GLA produced in our WT CHO cell line expressed as % of activity at 5 min after injection (n= 4). (c) Time-course analysis of GLA activities in plasma after infusion of Fabrazyme and engineered GLA variants as indicated expressed as % of activity at 5 min after injection (n=4 except for Fabrazyme in panel c where n=3).”

Query #11: On p11 the authors state that they picked out GLA-Bi23SA for animal experiments, based on its unique behavior. The authors should specify exactly what they mean with this statement, because, in the preceding paragraph (and in Fig 5f) they state and show that it behaves similar to the other glycoforms, except maybe in liver. Personally, I don't see much difference between GLA-Bi23SA and Fabrazyme (maybe the micrographs are too small). On the other hand, the measured activities (Fig 5e) do not always seem to agree with the tissue slices. Here, contrary to the impression I get from the slides, GL-Bi23SA shows significantly higher activity in the heart. In kidney, much lower activity than Fabrazyme, but on the micrograph, it looks the same to me. On the other hand, GLA-Bi26SA seems to have a more unique behavior to me, and opposite to GL-Bi23SA. Given the fact that Fabrazyme and GLA-Bi23SA don't show any difference in clearance of Gb3 from heart, it would be interesting to see if e.g. GLA-Bi26SA really results in higher Gb3 levels, or that the glycoform doesn't really matter.

Response #11: The statement refers primarily to the significantly longer circulation time (Fig. 5). We agree that the IHC show similar organ and cell distributions (apart from GLA-26SA in liver), which with the limitations of this assay is in agreement with all other presented data. We have now also included additional studies of GLA-26SA that confirms our previous conclusion (Fig. 5g).

Action #11: We have included the following text in Results (p. 12) to clarify this:

“Encouraged by the unique performances of GLA-Bi23SA with substantial extended circulatory half-life, and GLA-26SA with markedly altered biodistribution, we proceeded to test the effect of these two glycoforms on reduction of accumulated globotriosylceramide (Gb3) substrate in organs 2 wk after a single injection of 1 mg/kg. We performed two independent series with a single 1 mg/kg dose (Exp. #1) of Fabrazyme and GLA-Bi23SA and subsequently GLA-26SA (**Fig. 5g**). GLA-Bi23SA produced the same reduction of the Gb3 content in heart, kidney and liver as compared to Fabrazyme (**Fig. 5g**), and this corresponds to the greatest reduction reported so far with any enzyme strategy used in the Fabry mouse model^{22, 23, 65}. GLA-26SA produced lower reduction of Gb3 levels in heart and kidney compared to Fabrazyme and GLA-Bi23SA, while the effect in liver was similar for all variants. This finding correlates with the lower level of GLA-26SA distributed to heart and especially kidney (**Fig. 5e**).”

Query #12: Given the very low stability in plasma (half life of 15 or 45 mins?), it is surprising the authors can measure enzyme activity after 24h in the different organs. Could the authors discuss this and also why they chose to measure enzyme activities in the organs and biodistribution after 24h and Gb3 clearance only after 2 weeks? Would any differences not be much more pronounced if the measurements were done shortly after administering the enzymes? Circulation From Figs 5 b and c it is clear that most of the enzyme has been cleared from the circulation after 2 hours. One would think the biodistribution would be more informative at such a time interval than after 24 h, when clearance from/breakdown in an organ is maybe also playing an important role? Concerning Gb3 clearance, do the authors have an idea when the Gb3 levels reach a minimum value? Is this minutes, hours, days or weeks after administration?

Response #12: We disagree. The study design was selected to be as similar as possible to reports on studies with Fabrazyme in the field in order to make comparisons possible, which is indicated by references to appropriate prior studies (Current references 23, 38, 65, 76, and 77: Shen *et al.* 2016, Lee *et al.* 2003, Ioannou *et al.* 2001, Xu *et al.* 2015, and Benjamin *et al.* 2012). It is well established that GLA is rapidly taken up by cells, and that GLA after uptake is stable for 36-96 h in the lysosome (Ioannou *et al.* 2001 ; Current reference 65). Also analysis of biodistribution earlier than 24 h introduces potential issues with

contaminating enzyme activity in circulation. Moreover these previous studies (Ioannou *et al.* 2001; Current reference 65) showed that Gb3 will reaccumulate 3-4 wk after one injection, which indicate that the optimal time point for evaluation of Gb3 reduction is 1-2 wk.

In the revised Figure 5e we now include analysis of enzyme activity in organs after 1 wk with lower doses of Fabrazyme and GLA-Bi23SA clearly demonstrating detectable levels of enzyme above non-treated controls even at this time point in agreement with previous report (Xu *et al.* 2015; current reference 76).

Action #12: We refer to Response & Action to Query 5. We have also included the following sentence in the Online Methods section:

“All injections were performed via the tail-vein with enzymes diluted in saline to a total volume of 200 μ l per mouse, and the overall study design was identical to previous reports of studies with Fabrazyme and other GLA variants in this mouse model²³”

Query #13: More on protein clearance and uptake; How do protein clearance and uptake through lectins such as MR, AMR, MPR differ in man and mouse? I could imagine there might be differences, considering that normal mice proteins can have different terminating residues (alpha-gal, differences in sialic acid linkage, N-glycolyl neuraminic acid?). In other words, could the authors discuss how they think their findings in the mouse model would translate to human patients?

Response #13: To our knowledge there are no comparative studies demonstrating differences in binding specificities of murine and human MPR, AMR and MR, while differences have been shown for e.g. the macrophage galactose lectin MGL. Moreover, given the primary focus of the animal studies on the GLA-Bi23SA variant without M6P and exposed mannose we do not believe this is relevant.

Action #13: None

Reviewer #2

Query #1: According to the authors, their report is the first evidence that glycoforms capped with α 2-3 linked sialic acid but not α 2-6 sialic acid exhibit improved circulation in addition to other pharmacodynamic properties (page 4). However, an earlier report (Unverzagt *et al.*, J. Med. Chem. 2002, 45, 478) provided evidence that α 2-3 sialylated neoglycoproteins have longer serum half-life than their α 2-6 analogues. The authors should discuss their results within the context of Unverzagt and coworkers' earlier work.

Response #1: The statement p.4 is in principle correct in context, but we agree that without considering the context it may be misleading. We did cite Park *et al.* (PNAS 2005, Reference 58), a subsequent more detailed study also by Unverzagt, that presents the most complete and conclusive study of neoglycoproteins with α 2-6SA capping and AMR/ASGPR mediated clearance that in fact suggests its only α 2-6GalNAc β 1-4GlcNAc that is cleared. Since this is not found on CHO produced glycoproteins we do not think that a detailed discussion is warranted here.

Action #1: We have modified this statement p. 4 by specification as follows:

“We tested the performance of glycoforms without M6P and exposed Man residues to explore glycoforms without ligands for the major MPRs and MR receptors, and present the first evidence that GLA glycoforms capped with α 2-3 linked sialic acids (α 2-3SA), but surprisingly not α 2-6SA, exhibit improved circulation and biodistribution, and importantly with higher uptake in heart compared to the current leading agalsidase beta (Fabrazyme) ERT. Thus, in contrast to the current dogma, α 2-3SA capped glycoforms of at least some lysosomal enzymes may represent a new strategy to overcome the most critical problems of rapid clearance in liver and poor biodistribution found with current ERTs.”

We also modified the following text in the Results p.11:

“The striking increase in liver uptake of the α 2-6SA capped glycoform resembles previous studies obtained with albumin neoglyconjugates suggesting interaction of NeuAc α 2-6Gal β 1-4GlcNAc terminating glycans with the AMR⁶¹, although other studies have shown that it is primarily NeuAc α 2-6GalNAc β 1-4GlcNAc terminated and non-sialylated neoglycoproteins that are removed from circulation⁵⁸. Several therapeutic glycoproteins produced in human cells including HEK293 have partial α 2-6SA capping and appear to function similar to those produced in CHO cells with only α 2-3SA⁶², and further studies are needed to demonstrate how α 2-6SA capped glycoforms of ERTs and other therapeutic glycoproteins perform in humans. AMR-mediated uptake of α 2-6SA capped ERTs is predicted to be considerable less efficient compared to the asialo-glycoform based on previous studies demonstrating circulatory half-life of about 1 min for desialylated glucocerebrosidase compared to 21 min for the native enzyme⁶³.”

Query #2: Speculating on the mechanism of uptake of α 2-3SA capped glycoforms, the authors cited Markmann *et al.* work (reference 17) that lysosomal targeting of GLA is at least partly independent on M6P-tagging. The clause “at least partly” is confusing because Markmann *et al.* explicitly conclude that the targeting is M6P-dependent, and they did not restrict their findings to α 2-3SA capped glycoforms. The authors should address this.

Response #2: Markmann *et al.* (ref. 17) demonstrate that \approx 4% of GLA is targeted to the lysosome in mouse fibroblasts without M6P, and further demonstrates that two other lysosomal enzymes are targeted to the lysosome by LRP1. We discuss on p. 14 that megalin and sortilin has already been shown to serve in uptake and lysosomal targeting of GLA, but agree that further specification would clarify the text.

Action #2: The Discussion paragraph p.16 has been modified as follows:

“The mechanism for uptake of the α 2-3SA capped GLA glycoform is not clear, but studies have shown that lysosomal targeting of GLA is not exclusively dependent on M6P-tagging^{17, 73}, and endocytic receptors including sortilin (*SORT1*) and megalin (*LRP2*) that do not bind glycan features have been shown to serve in uptake of GLA^{74, 75}.”

Query #3: The authors failed to provide the rationale for selecting the five glycoforms tested in the in vivo experiments. Expanding the library of the tested glycoforms may provide a cue to the superior performance of the α 2-3SA, particularly the glycoforms without core fucose (Fig 3k, l and supplementary Fig 2 panel 60 and 61). Testing the non-fucosylated glycoforms may provide answers to the lower serum half-life of α 2-6SA compared with α 2-3SA given the reports of Unverzagt *et al.* (J. Med. Chem. 2002, 45, 478) that core fucosylation of a biantennary α 2,6-sialylated N-glycan significantly accelerates neoglycoprotein clearance from the bloodstream.

Response #3: We agree that the selection criteria for variants taken into animal studies should be stressed. We strongly disagree that there is a need to expand the glycoforms of GLA tested at this stage – as discussed above animal studies require important hypotheses and expectance of beneficial outcomes. We did not systematically address core fucosylation in the engineering because it anyway is glycosite specific and dependent on the N-glycan structures. Thus, e.g. the GLA-HiM6P variant has no core fucose (Fig. 5a), and there are no good comparative studies demonstrating potential positive effects on pharmacokinetic properties of natural glycoproteins.

Action #3: we have included the following text in the Results section p. 10:

“We then tested five distinct glycoforms of GLA, predicted to provide the broadest range of potential interactions with the major lectin receptors, M6PR, MR, and AMR, known to interact with and affect uptake and circulation of glycoproteins in blood (Fig. 5a)”

Query #4: It will be interesting to address the microheterogeneity observed in α 2-6SA glycoform (supplementary Fig 2, panel 58) and how this can affect clearance from the bloodstream (please see J. Med. Chem. 2002, 45, 478).

Response #4: While this would perhaps be academically interesting this is not straightforward given that the heterogeneity is mainly based on degree of α 2-6SA capping. For unknown reasons the α 2-3SA capping by KI of ST3GAL4 was much more effective than the α 2-6SA capping by KI of ST6GAL1 in contrast to our experience with other types of glycoproteins including EPO and IgG (Yang et al. 2015; reference 34 and Schulz et al. *Glycobiology* 2018). Regardless, the ASGPR-mediated clearance expected for exposed Gal residues would work in the same direction, albeit presumably less effective, as the α 2-6SA capping suggesting that this would not provide be beneficial.

Action #4: None

Query #5: Some studies show the superior pharmacokinetic performance of the α 2-6SA glycoform relative to the α 2-3SA (please see the review, Tejwani *et al.* *Biotechnol. J.* 2018, 13, 1700234, and the relevant reference therein). It will, therefore, be informative to test the effect of this α 2-6SA glycoform on the reduction of accumulated globotriaosylceramide (page 11) as they did for the α 2-3SA analogue. The results may substantiate their conclusion that uptake is independent of M6P and Man receptors as they alleged.

Response #5: We disagree. There are no studies showing superior pharmacokinetic performance of α 2-6SA capped glycoproteins or neoglycoproteins compared to α 2-3SA capping. There are multiple studies as referred to in the review pointed out by the Reviewer that demonstrates some differences in pharmacokinetics of the unique IgG molecules when capped by α 2-6SA, but this is based on comparison with the normal poor galactosylation and sialylation status of IgG and not α 2-3SA capping. We agree that analysis of the effect of the GLA-26SA variant on Gb3 clearance is relevant, and we now include additional animal studies.

Action#5: Results on Gb3 clearance with one 1 mg/kg dose of GLA-26SA after 2 wk is presented in the new Figure 5g, which shows predicted lower efficacy in reduction of Gb3 especially in the heart.

Minor points:

Page 3: “N-Acetyl-glucosamine (GlcNAc) and N-Acetyl-galactosamine (GalNAc) are typical written as “N-acetylglucosamine” and “N-acetylgalactosamine”

Page 10 (and maybe elsewhere): “mgs/L” should be “mg/L”

Page 15 (and elsewhere): the standard abbreviation for hours is “h” (not hrs)

Page 15, last paragraph: should there be a comma after “For GLA” at the start (also in this paragraph “ml” is used (“mL” is mostly used elsewhere, please be consistent)

Page 17 – heading of “Data Analysis” – based on usual formatting this would be “Data analysis”

References – check for consistent formatting (e.g., References 6, 17, 29 [and others] use upper case throughout the article title)

Response to the Minor points: Thank you, all fixed except we now use N-Acetyl-D-glucosamine (GlcNAc) and N-Acetyl-D-galactosamine (GalNAc) as suggested by SNFG.

REVIEWERS' COMMENTS:

Reviewer #1 (Remarks to the Author):

The authors discuss a comprehensive panel of CHO glycosylation mutant cell lines. Generating and characterizing these cell lines represents an impressive effort. The panel will prove to be an extremely valuable resource for any researcher interested in the effects of N-glycans on recombinant glycoproteins (also for other proteins than lysosomal enzymes, used here).

Perhaps even more importantly and certainly surprising: the authors also show that M6P N-glycans are not necessary for lysosomal targeting *in vivo*, in mice. This is a novel finding, which goes against the current view but it is convincingly supported by the experiments presented in this manuscript. It remains to be seen how this finding will translate to man, but the current study teaches us that one should keep an open mind and be wary of dogma. It might have far reaching consequences for ERT in lysosomal storage disease patients in the coming years. It is great to see researchers making this kind of effort. For that: my congratulations!

All my previously raised issues and questions have been thoroughly addressed. I have no further comments and I recommend publication of this excellent work.

Leander Meuris

Reviewer #2 (Remarks to the Author):

The authors satisfactorily addressed the concerns I raised in the original review